# A moisture-tolerant route to unprotected α/β-amino acid *N*-carboxyanhydrides and facile synthesis of hyperbranched polypeptides

Zi-You Tian[1], Zhengchu Zhang[1], Shuo Wang[1] & Hua Lu [1✉]

A great hurdle in the production of synthetic polypeptides lies in the access of *N*-carboxyanhydrides (NCA) monomers, which requires dry solvents, Schlenk line/gloveboxe, and protection of side-chain functional groups. Here we report a robust method for preparing unprotected NCA monomers in air and under moisture. The method employs epoxy compounds as ultra-fast scavengers of hydrogen chloride to allow assisted ring-closure and prevent NCA from acid-catalyzed decomposition under moist conditions. The broad scope and functional group tolerance of the method are demonstrated by the facile synthesis of over 30 different α/β-amino acid NCAs, including many otherwise inaccessible compounds with reactive functional groups, at high yield, high purity, and up to decagram scales. The utility of the method and the unprotected NCAs is demonstrated by the facile synthesis of two water-soluble polypeptides that are promising candidates for drug delivery and protein modification. Overall, our strategy holds great potential for facilitating the synthesis of NCA and expanding the industrial application of synthetic polypeptides.

[1] Beijing National Laboratory for Molecular Sciences, Center for Soft Matter Science and Engineering, Key Laboratory of Polymer Chemistry and Physics of Ministry of Education, College of Chemistry and Molecular Engineering, Peking University, Beijing 100871, People's Republic of China. ✉email: chemhualu@pku.edu.cn

Synthetic polypeptides, commonly prepared by the ring-opening polymerization (ROP) of amino acid N-carboxyanhydrides (NCA), are a family of biomimetic materials that can be applied as catalysts for heterogeneous reactions, self-assembling building blocks, surfactants for double emulsion, drugs carriers, tissue engineering scaffolds, antimicrobial agents, and alternatives to polyethylene glycol (PEG)[1–15]. For example, poly-L-alanine and poly-L-leucine are established industrial catalysts for asymmetric Juliá–Colonna epoxidation reactions[16]. Copaxone®, a random co-polypeptide prepared through the ROP of four different NCAs, is a blockbuster drug for multiple sclerosis which generated an annual global sales of 4 billion USD in 2012[17,18]. Several additional polypeptide-based therapeutic candidates are being tested at different phases of clinical trials[8,19]. Despite these achievements, synthetic polypeptides are difficult and costly to produce. Although recent advances in ROP has allowed, sometimes even in unpurified solvent or at aqueous-organic interfaces, the rapid generation of polypeptides with predictable and high molar masses ($M_n$), low dispersity (Đ), and well-defined reactive end groups[20–31], there are few new methods for simpler synthesis of NCA. The "Fuchs-Farthing's method" for the generation of NCA, which involves the phosgenation and ring-closure of amino acids via (tri)phosgene, is usually carried out under strictly anhydrous and air-free conditions (Fig. 1)[32]. Moreover, the traditional workup of NCA is notoriously tricky in that it requires redundant recrystallization in a glovebox using expensive dry solvents. Most importantly, although a few functional groups (e.g., alkene, alkyne, azido, ester, halogen, thioether, and selenoether) are tolerated on NCA[33–35], the introduction of more nucleophilic and/or proton-bearing functionalities such as alcohol, thiol, and carboxylic acid has been considerably difficult. Recently, Kramer and Deming achieved the notable breakthrough of purifying non-crystallizable NCAs via silica gel column chromatography in a glovebox[36]. In another outstanding study, Fuse reported efficient NCA synthesis in a mixture of organic and aqueous solvents with the use of a microflow reactor, followed by a sequence of flash dilution, extraction, and purification[37]. Despite all these advances, the scalable, open-vessel, and moisture-tolerant synthesis of challenging NCAs without the burden of protecting functional side groups remains an urgent and unmet need.

Here, we report a robust method for preparing unprotected NCA monomers in air and under moisture. The method employs epoxy compounds as ultra-fast scavengers of hydrogen chloride to allow assisted ring-closure and prevent NCA from acid-catalyzed decomposition under moist conditions (Fig. 1). The broad scope and functional group tolerance of the method are demonstrated by the facile synthesis of over 30 different α/β-amino acid NCAs, including many otherwise inaccessible compounds with reactive functional groups, at high yield and up to decagram scales. The utility of the method and the unprotected NCAs is demonstrated by the facile synthesis of two water-soluble polypeptides that are promising candidates for drug delivery and protein modification.

## Results

We began our current study by first seeking to identify the main hurdle of the generation of NCA in regular solvents (analytical grade) and without Schlenk line/glove box. The model monomer γ-benzyl L-glutamate NCA (Bn-GluNCA) was observed to undergo ROP and the reaction reached completion within 4 h in THF-$d_8$ (2.0 M) when stoichiometric $D_2O$ was added (~180,000 ppm, Fig. 2a and Supplementary Fig. 1). The conversion of Bn-GluNCA started out slow but then accelerated, indicating a relatively sluggish initiation by $D_2O$ compared to the amine-mediated chain propagation. In the presence of both DCl and $D_2O$, however, Bn-GluNCA was rapidly hydrolyzed instead of undergoing ROP, as evidenced by the precipitated amino acid and corresponding [1]H NMR (Fig. 2a and Supplementary Fig. 2). Therefore we concluded that HCl generated from the synthesis of NCA would cause the latter to rapidly decompose under hydrous conditions.

Several studies have demonstrated that the adverse effect of HCl can be minimized by employing an acid scavenger such as triethylamine (TEA), (+)-limonene, or α-pinene, particularly in the synthesis of challenging NCAs[38–40]. We ruled out the use of TEA because (i) it can prematurely initiate the ROP of NCA when in excess, (ii) it cannot quench the nucleophilic chloride, which is known to attack the carbonyls of NCAs to form isocyanates[41], and (iii) the removal of triethylammonium chloride salt adds extra burden to the workup procedure. On the other hand, we conducted the model reaction of α-pinene (1.0 M) and HCl (1.0 M) in THF-$d_8$ at room temperature, which showed low conversion and slow rate (Fig. 2b black curve and Supplementary Fig. 3). This is not surprising because α-pinene has been reported

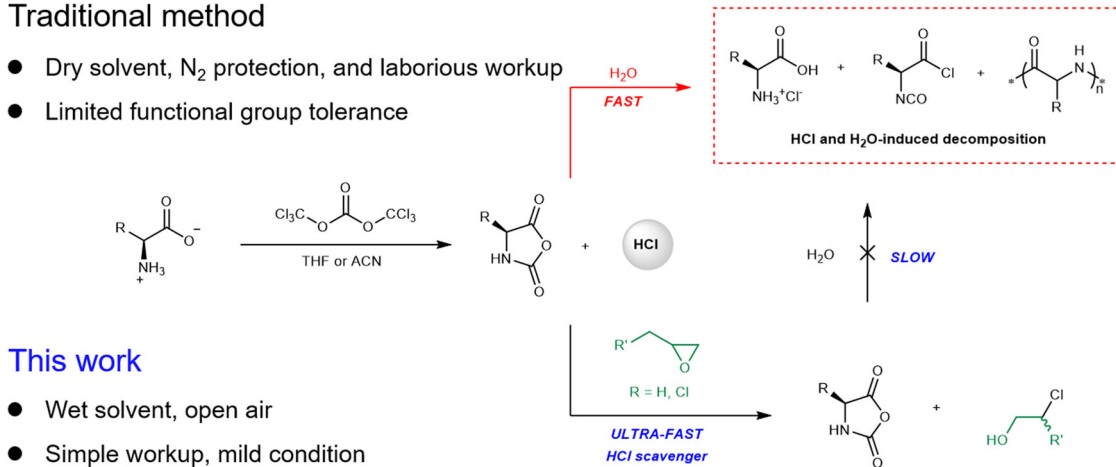

**Fig. 1 Comparison of traditional method and this work for the synthesis of NCAs.** This work reports the use of epoxy compounds as ultra-fast HCl scavenger for the moisture-tolerant synthesis of a broad scope of NCAs.

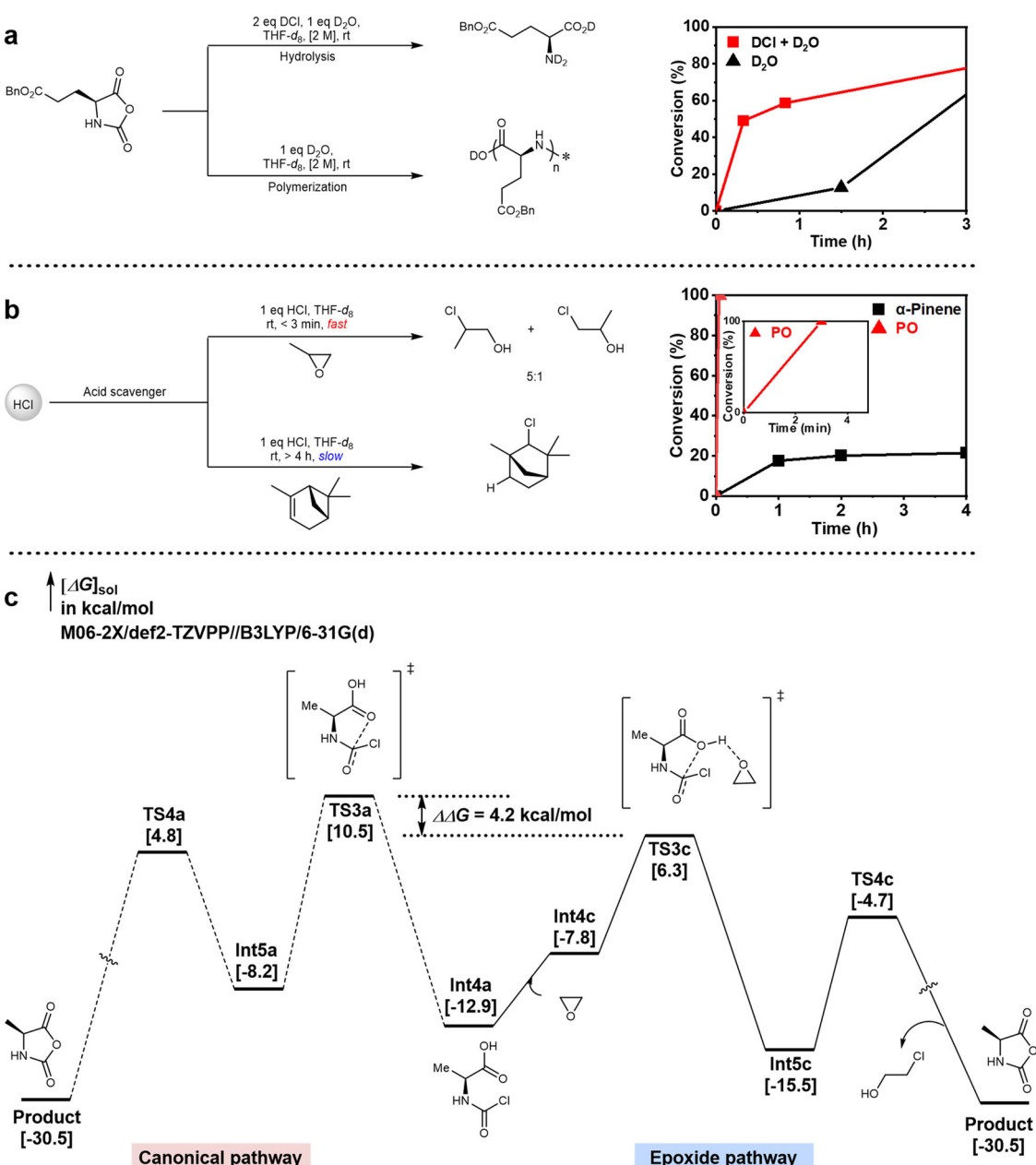

**Fig. 2 Mechanistic investigations and model reactions. a** Decomposition of Bn-GluNCA with $D_2O$ (black) or $D_2O/DCl$ (red). **b** Rate of HCl-scavenging by PO (red) or α-pinene (black) in THF-$d_8$ at room temperature. **c** DFT calculation of the ring-closure step in NCA synthesis with (right panel) or without (left panel) the assist of PO. (PO propylene oxide).

to react with acid at a much slower rate in THF compared to in other solvents such as chloroform and xylene[42]. Thus, a nonbasic HCl scavenger with greater efficiency than α-pinene was required.

We hypothesized that epoxide such as propylene oxide (PO) and epichlorohydrin (ECH) could be ideal additives in NCA synthesis, due to their ability to remove both proton and chloride even at relatively low temperatures (Fig. 2b)[43]. To our delight, the use of PO (1.0 M) led to complete, rapid conversion of HCl (1.0 M) in less than 3 min (Fig. 2b red curve) at room temperature, generating a mixture of 1-chloro-2-propanol and 2-chloro-1-propanol (Supplementary Fig. 4). We thus anticipated that PO could greatly facilitate the synthesis of NCA at room temperature, presumably because the acid-catalyzed epoxide ring-opening could generate sufficient heat. Indeed, density functional theory (DFT) calculations confirmed that ethylene oxide could lower the

energy barrier of the phosgenation reaction between L-alanine and phosgene by 4.2 kcal/mol, which implied a rate difference of ~1200-fold based on Curtin–Hammett principle and Eyring equation (Fig. 2c and Supplementary Figs. 5, 6).

We then reasoned that the NCA synthesis could be performed under moisture because (i) residual water in the solvent could be efficiently consumed by the excessive triphosgene, generating $CO_2$ and HCl, (ii) all hydrogen chloride could be rapidly removed by PO (Fig. 1), and (iii) PO, 1-chloro-2-propanol, and 2-chloro-1-propanol are all easy to remove during the workup. To test the idea, we mixed γ-benzyl L-glutamate (10 g, 1.0 equiv), triphosgene (0.5 equiv), and PO (4.0 equiv) in analytical grade THF (150 mL, $H_2O < 2000$ ppm) at room temperature in a sealed 350-mL heavy-wall flask. Unlike the conventional method, which would require heating the reaction mixture to 50–60 °C for 3–4 h under

**Fig. 3 Synthesis of NCAs with protected or unreactive side chains.** The addition of PO (propylene oxide) or ECH (epichlorohydrin) enables the facile synthesis of NCAs without the need of dry solvents, Schlenk line, or glove box.

nitrogen, our protocol achieved complete conversion of the amino acid substrate within ~1.5 h without external heating. Thanks to the efficient and complete elimination of HCl, pure Bn-GluNCA (Fig. 3, compound **3a**) could be obtained at 84% yield following recrystallization without the need for anhydrous solvents or a glovebox (environmental humidity: 70%). The NCA synthesis succeeded even with intentionally added $H_2O$ (1 equiv to an amino acid, >22,700 ppm) to the reaction mixture as long as triphosgene and PO were kept slightly excessive. The high purity of Bn-GluNCA was confirmed by a combination of $^1H$ and $^{13}C$ NMR spectroscopy, high-resolution mass spectrometry, and FT-IR spectroscopy (Supplementary Fig. 7). Increasing the loading of PO from 4 to 10 equivalent showed little effect on the yield (See Table S1). The method was also proved robust for the synthesis of many other NCAs (Fig. 3, compounds **3b**–**3o** and Supplementary Figs. 8–21), including two contained acid-labile protective groups that were difficult to handle previously. For glycine NCA (GlyNCA), ECH appeared to afford a faster reaction and higher yield than PO (Fig. 3, compound **3j** and Supplementary Fig. 16). Importantly, our method also exhibited satisfactory compatibility with normal column chromatography for the synthesis of non-crystallizable NCAs. One notable example was the hygroscopic γ-(2-(2-(2-methoxyethoxy)ethoxy)ethyl L-glutamate NCA (EG₃-GluNCA), which we were previously unable to synthesize at 40% environmental humidity and above, regardless of how other reaction conditions were optimized. In the current study, however, we successfully obtained 2.7 g of pure EG₃-GluNCA (Fig. 3, compound **3k** and Supplementary Fig. 17) with a separation yield of 72% by using analytical grade solvents (water <2300 ppm) and flash column chromatography under air (environmental humidity: 66%).

Pivoting to N-substituted NCAs (NNCA), we surprisingly ran into roadblocks in our initial attempts to prepare sarcosine NCA

(SarNCA) and ProNCA. Analysis of the crude reaction mixtures revealed the formation of byproducts from the aminolysis of PO by the secondary amines (Supplementary Fig. 22)[44]. We thus employed N-Boc amino acids as starting materials, which had been previously applied to NCA synthesis with success. To our gratification, we isolated SarNCA in 79% yield by conducting the reaction in an open flask at 0 °C to both ensure smooth gas release during the in situ deprotection of Boc and prevent the loss of PO vapor (Fig. 3, compound **4a** and Supplementary Fig. 23). The same strategy also proved successful in the synthesis of ProNCA (Fig. 3, compound **4b** and Supplementary Fig. 24) from N-Boc L-proline, attaining a purified yield of 72%, as compared to a previous protocol involving laborious and time-consuming workup and with only 30% overall yield[45]. Except for the hygroscopic EG₃-GluNCA that required extra caution for storage, all other purified NCAs in Fig. 3 were stored in −10 or −20 °C freezers without the need for desiccator or glove boxes (detailed information and shelf-life available in SI). The excellent polymerizability of the NCAs were confirmed by the ROP of Bn-GluNCA, EG₃-GluNCA, and SarNCA by using various initiators and under varying conditions, which showed good $M_n$ control, narrow Đ, and satisfactory reproducibility (See discussion of Supplementary Table 2 and Supplementary Fig. 25).

To further showcase the functional group tolerance of the method, we explored the generation of challenging NCAs that would otherwise be inaccessible. As reactive anhydrides, NCAs were thought to be incompatible with nucleophiles and/or reactive protons such as hydroxyl, amine, carboxylic acid, and thiol. Consistently, the synthesis of hydroxyl-bearing NCAs, such as L-serine NCA (SerNCA) and L-threonine NCA (ThrNCA)[46], had been attempted but without convincing characterization results. In contrast, we speculated that hydroxy-bearing NCAs should be well-tolerated by our method, based on the fact that neither 1-

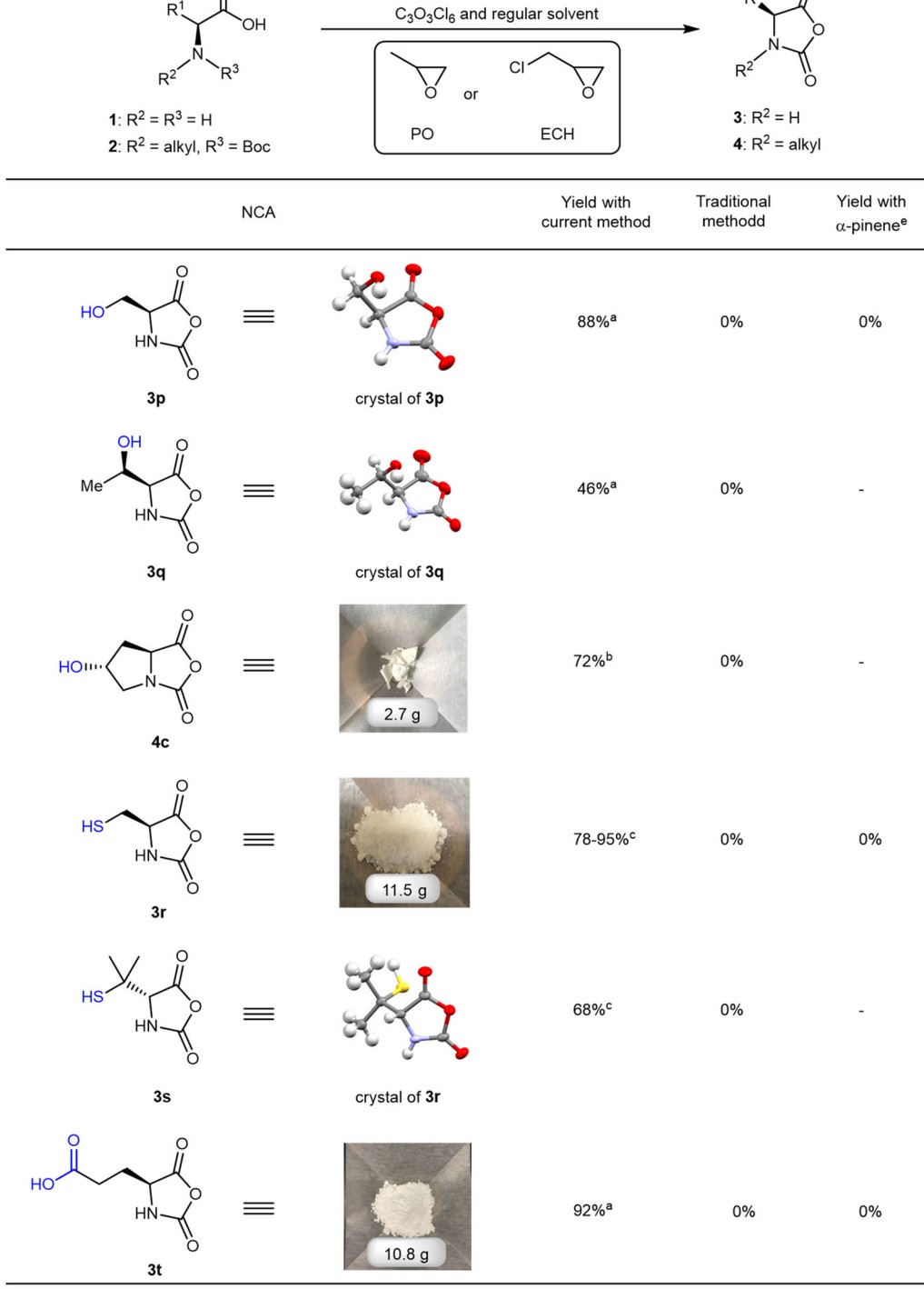

**Fig. 4 Synthesis of unprotected NCA.** The addition of PO (propylene oxide) or ECH (epichlorohydrin) enables the facile synthesis of challenging NCAs bearing reactive -OH, -SH, or -CO₂H side groups.

| | NCA | | Yield with current method | Traditional method[d] | Yield with α-pinene[e] |
|---|---|---|---|---|---|
| | **3p** | crystal of **3p** | 88%[a] | 0% | 0% |
| | **3q** | crystal of **3q** | 46%[a] | 0% | - |
| | **4c** | 2.7 g | 72%[b] | 0% | - |
| | **3r** | 11.5 g | 78-95%[c] | 0% | 0% |
| | **3s** | crystal of **3r** | 68%[c] | 0% | - |
| | **3t** | 10.8 g | 92%[a] | 0% | 0% |

[a]condition B: 0.5 eq triphosgene, 4 eq ECH, THF, rt.
[b]condition A-2: 0.5 eq triphosgene, 4 eq PO, CH₃CN, 0 °C.
[c]condition A-1: 0.5 eq triphosgene, 4 eq PO, THF, rt.
[d]condition: 0.5 eq triphosgene, anhydrous THF, 50 °C.
[e]condition: 0.5 eq triphosgene, 4 eq α-pinene, THF, rt.

chloro-2-propanol nor 2-chloro-1-propanol, the ring-opening products of PO and HCl, exhibited any detrimental effect on product formation. Indeed, SerNCA (compound **3p**, Fig. 4 and Supplementary Fig. 26) and ThrNCA (compound **3q**, Fig. 4 and

Supplementary Fig. 27) could be isolated at 71 and 46% yield (environmental humidity 68%), respectively. The moderate decrease in the yield of ThrNCA was not due to low substrate conversion, but rather, the rearrangement of the product forming

a 2-oxazolidone derivative during workup. As a further example, 4-hydroxy L-proline NCA (compound **4c**, HypNCA, Fig. 4 and Supplementary Fig. 28) was also obtained in 75% yield by using N-Boc 4-hydroxy L-proline as the starting material. As a strong nucleophile, free thiol has not been directly introduced to the side group of NCA before, even though a few papers claimed the generation of such NCAs but without concrete characterization[47]. To our great surprise, we successfully synthesized pure L-cysteine NCA (compound **3r**, CysNCA, Fig. 4 and Supplementary Fig. 29) bearing a primary thiol and D-penicillamine NCA (compound **3s**, PenNCA, Fig. 4 and Supplementary Fig. 30) with a tertiary thiol. The production of the temperature-sensitive CysNCA was accomplished in 78–94% yield via recrystallization (See SI) and could be easily scaled up to ~10 g per batch (see SI). The unprotected L-glutamic acid NCA (compound **3t**, GluNCA, Fig. 4 and Supplementary Fig. 31) was also smoothly prepared with ECH, but not with PO, and ~10 grams of the product was obtained after recrystallization at a yield of 92%. Again, it should be emphasized that all syntheses and workup procedures were conducted using analytical grade solvents and without nitrogen protection, and none of these NCAs was found to be attainable using a regular method or by adding α-pinene as an additive (0% purification yield, Fig. 4). All the NCAs prepared in Fig. 4 were stable in −20 °C freezers for one to several months without the need for an extra desiccation agent or apparatus.

As further proof that our unprotected NCAs hold tremendous synthetic value, the direct ROP of GluNCA (**3t**) mediated by benzylamine (BnNH$_2$) in dry DMF provided one-step access to poly-L-glutamic acid (PLG), a therapeutically relevant biodegradable carrier polymer being evaluated in phase 3 clinical trials[19], at >95% monomer conversions within 3 h in general (Fig. 5a). This concise synthesis avoided the burdensome deprotection of poly(γ-benzyl L-glutamate) with highly corrosive HBr, leading to substantial time saving and yield increase compared to the conventional multistep procedure (yield: 80% vs. 40%). Matrix-assisted laser desorption/ionization-time of flight (MALDI-TOF) mass spectrometry of a 20-mer PLG offered solid evidence to confirm the proposing polymer structure, which included the benzyl amide terminus (Supplementary Fig. 32). Increasing the feeding monomer-to-initiator (M/I) from 20/1 to 50/1 resulted in a clear shift of the unimodal peak to higher $M_n$ in aqueous SEC, suggesting good polymerization control (Fig. 5b). Aqueous SEC analysis of the two polymers gave an $M_n$ of 3.6 ± 0.1 and of 8.2 ± 0.2 kg/mol (each data were presented as average ± standard deviations from two independent ROP experiments). The integration of Glu/Bn in the [1]H NMR spectra of the two PLG polymers agreed with the SEC results on $M_n$ (Supplementary Fig. 33), respectively. In situ chain extension from the ROP solution of PLG$_{50}$ with EG$_3$-GluNCA (Fig. 5c) successfully yielded the desired block copolymer PLG-block-P(EG$_3$-Glu) as characterized by SEC (Fig. 5d) and [1]H NMR (Supplementary Fig. 34). Manipulating the topology and architecture of macromolecules has been an important approach toward tailored functions[48]. For example, branched polymers are promising materials in nanomedicine and self-assembly due to their unique physicochemical properties such as high degree of functional groups, three-dimensional architecture, enhanced solubility, less chain entanglement, and desirable rheological and lubricating properties[49,50]. Unfortunately, conventional access to branched polypeptides entails the protection of amino groups prior to polymerization and their subsequent deprotection to introduce the branching sites[51]. Here, we theorized that thiol could be used as an initiation group for the ROP of NCA[25] and CysNCA (**3r**) could serve as an inimer. Copolymerization of CysNCA (**3r**) and EG$_3$-GluNCA (**3k**) by BnNH$_2$/4-(N,N-dimethylamino)pyridine (DMAP) in dry DMF afforded P(EG$_3$-Glu)-

branch-PCys in a single step, which branches off at the thioester linkages (Fig. 5e). Based on circular dichroism (CD) spectroscopy, P(EG$_3$-Glu)-branch-PCys exhibited a characteristic α-helical conformation with a helicity of ~73% (Fig. 5f). DMF-phased SEC of P(EG$_3$-Glu)-branch-PCys found its $M_n$ and Đ to be 33 kg/mol and 1.66, respectively (Fig. 5g, red trace). Elemental analysis of P(EG$_3$-Glu)-branch-PCys gave a sulfur content of 2.4 wt%, and thus a molar EG$_3$-Glu/Cys ratio of ~4.8/1, which agreed well with the [1]H NMR spectroscopy (Supplementary Fig. 35). Ellman's assay of the two polymers suggested that at least 45% of all Cys still carried a free thiol. Treatment of the hyperbranched polymer with tris(2-carboxyethyl)phosphine (TCEP), a reductant for disulfide bond cleavage, did not alter the aqueous SEC profile, suggesting no disulfide bond formation (Supplementary Fig. 36). Interestingly, the polymer product was partially degraded when treated with cysteine in water, as evidenced by the increased elution time in SEC (Fig. 5g, blue trace). The degradation was likely caused by the native chemical ligation (NCL) reaction that cleaves the thioester branching sites (Fig. 5e). In another attempt, copolymerization of CysNCA (**3r**, 25 equiv) and Bn-GluNCA (**3a**, 50 equiv) by BnNH$_2$ (1.0 equiv)/DMAP (1.0 equiv) in dry DMF afforded PBLG-branch-Pcys in a similar fashion (Fig. 5e). DMF-phased SEC analysis of the copolymer gave an $M_n$ and Đ of 14.4 kg/mol and 1.27, respectively (Supplementary Fig. 37). The BnGlu/Cys molar ratios was determined as 3.3/1 according to [1]H NMR spectroscopy (Supplementary Fig. 38). Two dimensional NMR experiments (COSEY and HSQC, Supplementary Figs. 39, 40) revealed two sets of Cys peaks assignable to Cys in the form of free thiol (linear connection) and thioester (branching sites), respectively. Treatment of the polymer by benzyl mercaptane and TEA again led to partial degradation of the copolymer, further corroborating the existence of thioester linkages (Supplementary Fig. 37).

Building on our success with NCAs, we set out to expand the scope of our protocol to α-hydroxyl acid O-carboxyanhydrides (OCA)[52–54] and β-amino acid NCA (βNCA)[55]. Similar to that of NCA, the synthesis of OCA generally involves stringent anhydrous conditions, but also requires activated charcoal to remove impurities. Here, by implementing PO as the acid scavenger, the representative L-phenyllactic acid OCA (Fig. 6, compound **7a** and Supplementary Fig. 41) and L-mandelic OCA (Fig. 6, compound **7b** and Supplementary Fig. 42) were smoothly produced in modest yields (50–69%). Of note, the synthesis of OCA required a longer time than that of NCA for the reason that the former's hydroxy group is less nucleophilic than the latter's amino group. On the other hand, βNCAs, which are considerably more labile than α-amino acid NCAs, could also be conveniently synthesized in excellent yields (68–96%; Fig. 6, compounds **8a**–**f** and Supplementary Figs. 43–48) with ECH as an additive and under atmospheric conditions. Notably, βNCAs prepared by the flow chemistry method were reported to result in easy product decomposition during the workup or storage[55]. Here, the enhanced stability of βNCA suggested an absence of side reactions due to the complete removal of HCl, which once again underscored the power of our method.

## Discussion

Synthetic polypeptides have gained popularity as biodegradable and protein-mimicking polymers with broad application potential, but the preparation of the NCA monomers requires stringent moisture-free conditions and tricky workup procedures. The effect of HCl-removing in the formation of NCA has been partially recognized by the field, but the detrimental role of HCl has not been fully elucidated from a mechanistic point of view. Acid scavengers, such as TEA and α-pinene, have been used as

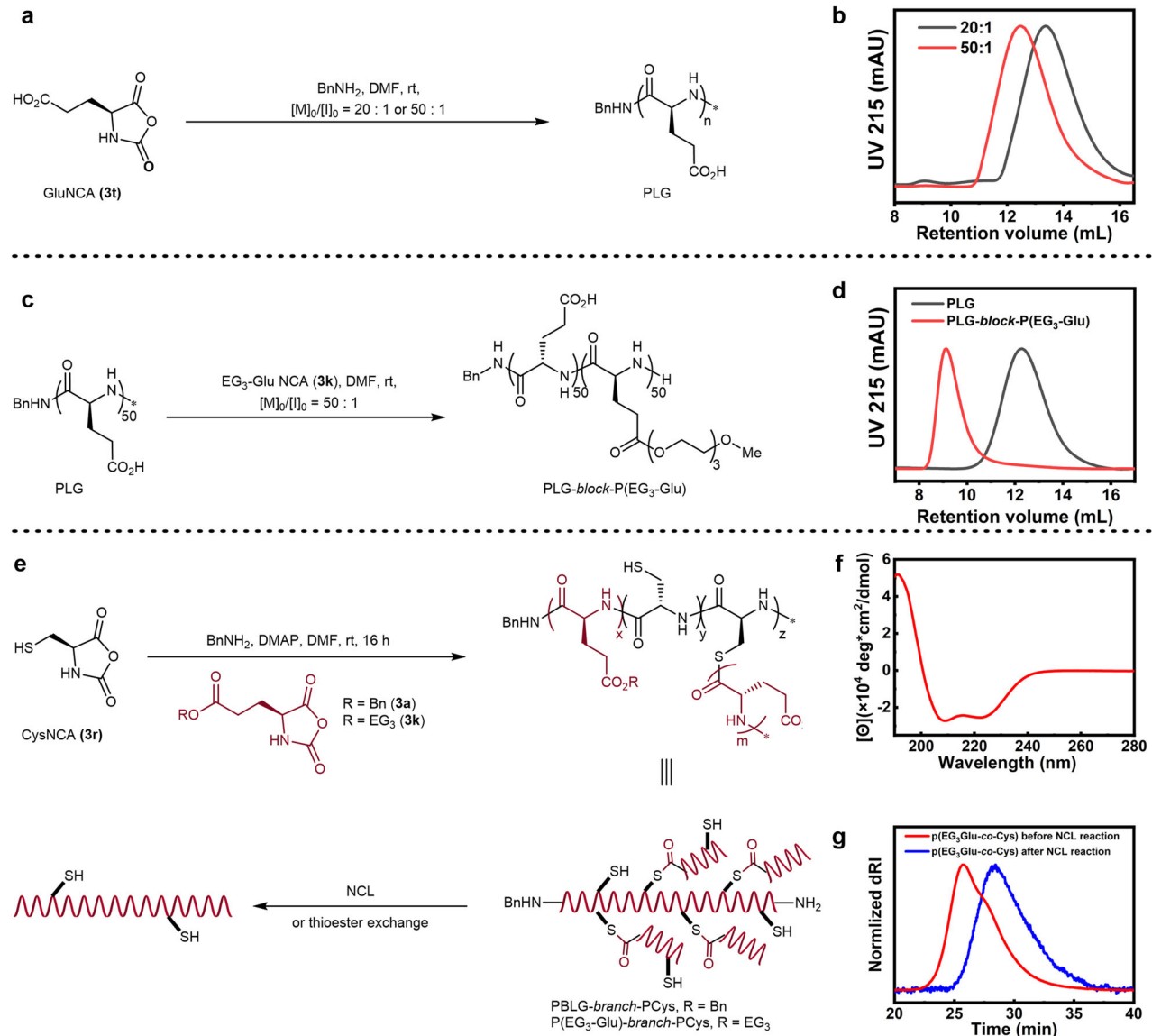

**Fig. 5 Homo- and co-ROP of GluNCA and CysNCA. a**, **b** Synthetic scheme (**a**) and SEC (**b**) characterization of PLG via the direct ROP of GluNCA at a feeding monomer/initiator ratio of 20/1(black trace) and 50/1 (red trace). The $M_n$ values reported here were obtained from a SEC equipped with a multi-angle light scattering detector and a refractive index detector. mobile phase: 50 mM $NaH_2PO_4$, pH = 9 in $H_2O$; flow rate of 0.5 mL/min; d$n$/d$c$ of PLG = 0.1184 ± 0.002859 mL/g). **c**, **d** Synthetic scheme (**c**) and SEC (**d**) characterization of the block polymer PLG-*block*-P(EG₃-Glu). **e**–**g** Synthetic scheme (**e**), CD (**f**), and SEC in DMF (**g**) characterization of P(EG₃-Glu)-*branch*-PCys (red trace) via the ROP of EG₃GluNCA and CysNCA. The blue trace in (**g**) represents the partially degraded polypeptides after native chemical ligation (NCL).

additives and achieved certain successes in the synthesis of some, but not all, NCA monomers. In Fig. 2a, we demonstrated that the stability of NCA towards water is highly dependent on the presence of HCl. We, therefore, propose that NCA synthesis under mild, atmospheric conditions, without the need for moisture removal, could be achievable as long as HCl could be efficiently removed from the crude products before the HCl-catalyzed NCA decomposition taking place. After ruling out TEA and α-pinene, we identified PO and ECH as ultra-fast and clean HCl-quenching reagents suitable for NCA preparation. All NCAs were comprehensively characterized and exhibited comparable polymerizability to those prepared by other methods. Thanks to the almost instantaneous and complete removal of HCl from the reaction mixture, which desensitized the monomers to moisture, some of the NCAs prepared by this method showed improved storage convenience with no need of a desiccator or glovebox (See SI). In

addition to the above-mentioned benefits, the rapid removal of HCl by PO/ECH also allowed accelerated preparation of NCA via lowering the cyclization energy barrier (Fig. 2c), which is important for the synthesis of some labile or highly reactive monomers without heating. The subsequent investigations showed that our method could be generally applied to a broad scope of substrates, and could easily be executed on a decagram scale to produce challenging NCAs that carry highly reactive –OH, -COOH, and –SH groups (Fig. 4). The use of unprotected NCAs can improve the efficiency of polypeptide synthesis by obviating the need for tedious protection and deprotection. This is vividly demonstrated through the direct ROP of GluNCA (**3t**) for the preparation of PLG with a much shorter procedure and spared efforts in protection and deprotection (Fig. 5a). In opposition to the intuition of a significantly reduced rate, surprisingly, the rate of ROP of GluNCA at low M/I ratios of 20/1–50/1

**Fig. 6 Synthesis of βNCAs and OCAs.** The addition of PO (propylene oxide) or ECH (epichlorohydrin) enables the facile synthesis of βNCAs and OCAs without the need of dry solvents, Schlenk line, or glove box.

appeared not affected significantly by the presence of free carboxylic acid in the side chain when the monomer concentration was sufficiently high. The mild condition, good polymerization control, high atom economy, and time-saving procedure cannot be underrated from the perspective of cost reduction, scale-up, and sustainability. Apart from homopolymerizations, copolymerizations of the unprotected NCAs would also be highly useful in producing functional polypeptides (Fig. 5c, e). For example, random copolymerization of CysNCA (**3r**) with other NCAs allowed facile production of branched polypeptides, further underscoring the synthetic utility of unprotected NCAs, which were attainable only with the epoxy-assisted NCA synthesis. We have previously shown that the linear helical P(EG$_3$-Glu) is a promising non-fouling polypeptide for surface coating and

modification of protein drugs[5,6,56–59]. We envision the broader application of P(EG$_3$-Glu)-*branch*-PCys as a high-performance biomedical material owing to its unique topology and the intrinsically embedded reactive thiol and thioester functionalities, which is amenable to further bioconjugation, surface anchoring, and responsive degradation. It should be pointed out that the results of the copolymerization could be modulated by simply altering the feeding ratio, allowing fine-tuning of $M_n$ and microstructure of the polymer products. Moreover, the branching degree of the copolymer can be potentially manipulated via intra/inter-molecular *S*-to-*N* acyl shift during the course of polymerization. Detailed kinetic studies, mechanistic investigations, and optimization of the homopolymerization of GluNCA at higher M/I ratios, copolymerization of CysNCA with other

NCAs, along with the related application testing are currently underway in our laboratory. The generality of the method was further demonstrated by the synthesis of several OCAs and βNCAs in high purity (Fig. 6), paving the road for easier access to various functional poly(α-hydroxyl acid)s and β-polypeptides. Last but not least, we expect the method detailed in this study to be potentially adaptable to industrial-scale production for reasons that (i) both PO and ECH are relatively inexpensive and easily available, (ii) the reactions can be performed without the need for ultra-dry solvents, nitrogen protection, or glovebox, leading to considerably simpler procedures and cost reduction; (iii) the instant removal of HCl can prevent the erosion of equipment, and (iv) the solvents, PO, ECH, and the ring-opening byproducts can all be easily recycled through distillation. Taken together, the simplicity, robustness, and scalability of our method promises may strongly facilitate the NCA synthesis and the popularization of synthetic polypeptides in both laboratories and industry, and open up numerous possibilities for the development of high-performance polypeptide materials.

## Methods

**Synthesis of 3a (Bn-GluNCA).** To a pressure vessel with a heavy wall (air humidity: 70%), γ-benzyl L-glutamate **1a** (10.0 g, 42.1 mmol, 1.0 eq), THF (150 mL), methyloxirane (13.0 mL, 169 mmol, 4.0 eq) were added sequentially under magnetic stirring. Triphosgene (6.3 g, 21.1 mmol, 0.5 eq) was finally added in one portion and the vessel was sealed immediately. The amino acid gradually disappeared in ~30 min with noticeable heat release. The reaction was stirred at room temperature for ~1.5 h in total and cooled down to ~4 °C in an ice bath. For safety reasons, the excessive triphosgene was quenched by adding 70 mL cold water at ~4 °C with 1–3 min stirring. The mixture was extracted with ethyl acetate (EA, 50 mL × 2) at room temperature. The combined organic phase was washed with brine and dried with anhydrous $Na_2SO_4$. After the removal of the solvent by rotatory evaporation in a vacuum under 45 °C, the crude product was purified by crystallization in hexane/THF below 10 °C (preferably in a cold room) without $N_2$ protection. The pure product **3a** was obtained as a white needle crystal (9.3 g, yield = 84%). The NCA was stored at −10 °C for at least 5 months.

$^1$**H NMR** (400 MHz, CDCl$_3$) δ 7.42 – 7.31 (m, 5H), 6.70 (br, 1H), 5.13 (s, 2H), 4.38 (ddd, J = 6.7, 5.4, 1.0 Hz, 1H), 2.59 (t, J = 6.9 Hz, 1H), 2.32 – 2.21 (m, 1H), 2.18 – 2.06 (m, 1H).

$^{13}$**C NMR** (101 MHz, CDCl$_3$) δ 172.5, 169.6, 152.2, 135.3, 128.8, 128.7, 128.4, 67.2, 57.0, 29.8, 26.9.

**HRMS** (ESI − FTICR, *m/z*): [M + H]$^+$ calculated for $C_{13}H_{14}NO_5^+$: 264.0866; found: 264.0865.

**FT-IR** (cm$^{-1}$) 1854, 1781, 1728.

**Synthesis of 3k (EG$_3$-GluNCA).** To a pressure vessel with a heavy wall (air humidity: 66%), γ-(2-(2-(2-methoxyethoxy)ethoxy)ethyl L-glutamate **1k** (2.8 g, 9.6 mmol, 1.0 eq), THF (50 mL), methyloxirane (6.7 mL, 96.1 mmol, 10.0 eq) were added sequentially under magnetic stirring. Triphosgene (1.5 g, 4.8 mmol, 0.5 eq) was finally added in one portion and the vessel was sealed immediately. The reaction was stirred at room temperature for 2 h. (This NCA cannot be washed with water!!!) After the removal of the solvent by rotatory evaporation in a vacuum under 50 °C, the crude product was purified by flash column chromatography (from PE/EA = 5:1 gradually to pure EA). The pure product **3k** was obtained as a colorless oil (2.2 g, yield = 72%). The NCA was stored at −30 °C for 1 month.

$^1$**H NMR** (400 MHz, DMSO-*d*$_6$) δ 9.09 (br, 1H), 4.46 (ddd, J = 7.9, 5.4, 1.2 Hz, 1H), 4.20 – 4.07 (m, 2H), 3.60 (dd, J = 5.3, 4.2 Hz, 2H), 3.57 – 3.47 (m, 5H), 3.46 – 3.39 (m, 2H), 3.24 (s, 3H), 2.47 (t, J = 7.6 Hz, 2H), 2.10 – 1.96 (m, 1H), 1.99 – 1.83 (m, 1H).

$^{13}$**C NMR** (101 MHz, DMSO-*d*$_6$) δ 171.8, 171.4, 151.9, 71.3, 69.8, 69.7, 69.6, 68.2, 63.5, 58.1, 56.2, 29.1, 26.5.

**HRMS** (ESI − FTICR, *m/z*): [M + H]$^+$ calculated for $C_{13}H_{22}NO_8^+$: 320.1340; found: 320.1339.

**FT-IR** (cm$^{-1}$) 1855, 1782.

**Synthesis of 3r (CysNCA).** To a pressure vessel with a heavy wall (air humidity: 50%), L-cysteine **1r** (10.0 g, 82.5 mmol, 1.0 eq), THF (200 mL), methyloxirane (23 mL, 330.2 mmol, 4.0 eq) were added sequentially under magnetic stirring. Triphosgene (12.3 g, 41.3 mmol, 0.5 eq) was finally added in one portion and the vessel was sealed immediately. The reaction was stirred at room temperature for 7.5 h and filtered directly. (This NCA cannot be washed with water!!!) The filtrate was concentrated by rotatory evaporation in a vacuum under 40 °C and during the course, the crude product precipitated out as a crystal with ~1–2 mL oil left in the flask. The wet crude product was then added a mixture of PE/EA (10/1, 75 mL

total), stirred, sit for 1 min, and the liquid phase was decanted. The washing was repeated three times at room temperature with no need of $N_2$ protection. The pure product **3r** was obtained as a white crystal (multiple attempts, 9.4–11.5 g, yield = 78–95%). The NCA was stored at −20 °C for 1 month. For longer storage, the NCA can be stored in a glove box at −30 °C.

$^1$**H NMR** (400 MHz, THF-*d*$_8$) δ 8.00 (br, 1H), 4.56 (ddd, J = 5.3, 4.1, 1.2 Hz, 1H), 3.98 – 2.82 (m, 2H), 2.05 (dd, J = 9.6, 8.3 Hz, 1H).

$^{13}$**C NMR** (101 MHz, THF-*d*$_8$) δ 170.1, 152.9, 60.5, 26.5.

**HRMS** (ESI − FTICR, *m/z*): [M + H]$^+$ calculated for $C_4H_6NO_3S^+$: 148.0063; found: 148.0064.

**FT-IR** (cm$^{-1}$) 1863, 1792.

**Synthesis of 3t (GluNCA).** To a pressure vessel with a heavy wall (air humidity: 40%), L-glutamic acid **1t** (10.0 g, 68.0 mmol, 1.0 eq), THF (150 mL), ECH (16.0 mL, 203.9 mmol, 4.0 eq) were added sequentially under magnetic stirring. Triphosgene (10.0 g, 34.0 mmol, 0.5 eq) was finally added in one portion and the vessel was sealed immediately. The reaction was stirred at room temperature for 18 h and filtered directly. After the removal of the solvent by rotatory evaporation in a vacuum under 45 °C, the crude product was purified by crystallization in PE/THF below 10 °C (preferably in a cold room) without $N_2$ protection. The pure product **3t** was obtained as a white powder (10.8 g, yield = 92%). The NCA was stored at −20 °C for 2 months.

$^1$**H NMR** $^1$H NMR (400 MHz, DMSO-*d*$_6$) δ 12.32 (br, 1H), 9.09 (br, 1H), 4.45 (t, J = 7.1 Hz, 1H), 2.36 (t, J = 7.1 Hz, 2H), 1.99 (dq, J = 14.1, 7.1 Hz, 1H), 1.85 (dq, J = 14.1, 7.1 Hz, 1H).

$^{13}$**C NMR** (101 MHz, DMSO-*d*$_6$) δ 173.4, 171.5, 152.0, 56.3, 29.1, 26.6.

**HRMS** (ESI − FTICR, *m/z*): [M + H]$^+$ calculated for $C_6H_8NO_5^+$: 174.0397; found: 174.0395.

**FT-IR** (cm$^{-1}$) 1858, 1791, 1741.

**Preparation of poly-L-glutamic acid (M/I = 20/1).** To the solution of L-glutamic acid NCA (250 mg, 1.4 mmol, 20 eq) in anhydrous DMF (500 μL), the initiator benzylamine (72 μL × 1.0 M, 1.0 eq) was added. The reaction was stirred at room temperature for 16 h. After extensive dialysis and lyophilization, the polymer was obtained as a white powder (138 mg, yield = 74%). The experiments were independently repeated twice. The obtained $M_n$ was 3.6 ± 0.1 kg/mol, and Đ was in the range of 1.06–1.08 determined by aqueous SEC. The d$n$/d$c$ value of PLG in PBS was measured as 0.1184 ± 0.002859 mL/g (See supplementary information).

**Preparation of poly(L-glutamic acid)-*block*-(EG$_3$-Glu).** To the solution of L-glutamic acid NCA (250 mg, 1.4 mmol, 50 eq) in dry DMF (0.5 mL), the initiator benzylamine (29 μL × 1.0 M, 1.0 eq) was added. The reaction was stirred at room temperature for 24 h. Then, 54 μL (58 mM) of the solution of poly(L-glutamic acid) was added to the solution of EG$_3$-GluNCA (50 mg, 0.16 mmol) in dry DMF (300 μL). The reaction was stirred in the glove box at room temperature for 24 h. After extensive dialysis and lyophilization, the polymer was obtained as a white fiber solid.

**Synthesis of 4b (ProNCA).** To a 100 mL flask (air humidity: 74%), Boc-L-proline **2b** (1.0 g, 4.7 mmol, 1.0 eq), CH$_3$CN (15 mL), methyloxirane (3.7 mL, 47 mmol, 10.0 eq) were added sequentially under magnetic stirring. Triphosgene (700 mg, 2.3 mmol, 0.5 eq) was finally added in one portion and the flask was kept slightly open to air to allow $CO_2$ escape. The reaction was stirred at ~4 °C in an ice bath for 2.5 h. For safety reasons, the excessive triphosgene was quenched by adding 0.5 mL cold water at ~4 °C with 1–3 min stirring. Then brine (1 mL) was added to facilitate phase separation and subsequent extraction. The aqueous phase was extracted with ethyl acetate (EA, 10 mL × 2). The combined organic phase was directly dried with anhydrous $Na_2SO_4$. After the removal of the solvent by rotatory evaporation in a vacuum under 40 °C, the crude product was purified by flash column chromatography (from PE/EA = 5:1 to 1:1 to pure EA) as soon as possible. The pure product **4b** was obtained as light yellow oil, which became a solid after frozen (470 mg, yield = 72%). The NCA was stored at −20 °C for 1 month. (To obtain a higher yield, recrystallization of Boc-L-proline prior to use is highly recommended.)

$^1$**H NMR** (400 MHz, CDCl$_3$) δ 4.32 (dd, J = 9.1, 7.4 Hz, 1H), 3.73 (dt, J = 11.3, 7.4 Hz, 1H), 3.30 (ddd, J = 11.3, 8.5, 4.6 Hz, 1H), 2.28 (dtd, J = 12.4, 7.4, 3.7 Hz, 1H), 2.24 – 2.14 (m, 1H), 2.09 (dtd, J = 14.3, 9.1, 7.4, 4.6 Hz, 1H), 1.92 (dq, J = 12.4, 9.1 Hz, 1H).

$^{13}$**C NMR** (101 MHz, CDCl$_3$) δ 168.9, 154.9, 63.1, 46.5, 27.6, 26.9.

**HRMS** (ESI − FTICR, *m/z*): [M + H]$^+$ calculated for $C_6H_8NO_3^+$: 142.0499; found: 142.0499.

**FT-IR** (cm$^{-1}$) 1848, 1763.

**Synthesis of 4c (HypNCA).** To a 100 mL flask (air humidity: 72%), Boc-L-hydroxyproline **2c** (5.0 g, 21.6 mmol, 1.0 eq), CH$_3$CN (50 mL), methyloxirane (15.1 mL, 216 mmol, 10.0 eq) were added sequentially under magnetic stirring. Triphosgene (3.222 g, 10.8 mmol, 0.5 eq) was finally added in one portion and the flask was kept slightly open to air to allow $CO_2$ escape. The reaction was stirred at ~ 4 °C in an ice bath for 2.5 h. For safety reasons, the excessive triphosgene was

quenched by adding 0.5 mL cold water at ~4 °C with 1–3 min stirring. Then 1 mL brine was added to separate the monophase. The organic phase was directly dried with anhydrous $Na_2SO_4$. After the removal of the solvent by rotatory evaporation in a vacuum under 35 °C (Do not rotavapor too long), the crude product was purified by flash column chromatography (from PE/EA = 5:1 to 1:1 to pure EA). The rude product **4c** was obtained as a colorless oil at room temperature with ~5% solvent residue (2.7 g, yield = 75%). After being stored in a −20 °C freezer, the oil became a chunk-like solid. Purer HypNCA was attainable via careful recrystallization of the crude product in mixed THF/hexane. The NCA was stored at −20 °C for 1 month. (recrystallization of Boc-Hyp prior to use is highly recommended).

**¹H NMR** (400 MHz, DMSO-$d_6$) δ 5.31 (br, 1H), 4.72 (dd, J = 10.8, 6.8 Hz, 1H), 4.54 (ddd, J = 4.8, 4.8, 1.5 Hz, 1H), 3.71 (dd, J = 11.2, 4.8 Hz, 1H), 3.06 (dd, J = 11.2, 1.5 Hz, 1H), 2.23 (ddd, J = 12.2, 10.8, 4.8 Hz, 1H), 1.95 (dd, J = 12.2, 6.8 Hz, 1H).

**¹³C NMR** (101 MHz, DMSO-$d_6$) δ 170.3, 154.7, 72.6, 62.3, 55.0, 35.5.

**HRMS** (ESI − FTICR, m/z): $[M + H]^+$ calculated for $C_6H_8NO_4^+$: 158.0448; found: 158.0447.

**FT-IR** (cm⁻¹) 3419, 1851, 1760.

**Synthesis of 7a (L-phenyllactic acid OCA).** To a pressure vessel with a heavy wall (air humidity: 55%), L-phenyllactic acid **5a** (1.0 g, 6.0 mmol, 1.0 eq), THF (15 mL), methyloxirane (4.2 mL, 60 mmol, 10.0 eq) were added sequentially under magnetic stirring. Triphosgene (893 mg, 3.0 mmol, 0.5 eq) was finally added in one portion and the vessel was sealed immediately. The reaction was stirred at room temperature for 24 h in total. After the removal of the solvent by rotatory evaporation in a vacuum under 45 °C, the crude product was purified by crystallization in hexane/THF below 10 °C (preferably in a cold room) without $N_2$ protection. The pure product compound **7a** was obtained as a white needle crystal (750 mg, yield = 65%). The NCA was stored at −10 °C for 1 month.

**¹H NMR** (400 MHz, CDCl₃) δ 7.41 − 7.29 (m, 3H), 7.26 − 7.19 (m, 2H), 5.30 (t, J = 4.9 Hz, 1H), 3.38 (dd, J = 14.9, 4.9 Hz, 1H), 3.25 (dd, J = 14.9, 4.9 Hz, 1H).

**¹³C NMR** (101 MHz, CDCl₃) δ 166.5, 147.9, 131.6, 129.8, 129.3, 128.5, 80.0, 36.5.

**HRMS** (EI, m/z): $[M]^+$ calculated for $C_{10}H_8O_4^+$: 192.04171; found: 192.04173. FT-IR (cm⁻¹) 3451, 1883, 1792.

**Synthesis of 8c (L-aspartic acid β-methyl ester NCA).** To a pressure vessel with a heavy wall (air humidity: 80%), L-aspartic acid β-methyl ester **6c** (1.0 g, 6.9 mmol, 1.0 eq), THF (30 mL), ECH (2.1 mL, 27.6 mmol, 4.0 eq) were added sequentially under magnetic stirring. Triphosgene (1.0 g, 3.5 mmol, 0.5 eq) was finally added in one portion and the vessel was sealed immediately. The reaction was stirred at room temperature for 2 h. After the removal of the solvent by rotatory evaporation in a vacuum under 45 °C, the crude product was purified by crystallization in PE/ THF below 10 °C (preferably in a cold room) without $N_2$ protection. The pure product **8c** was obtained as a white crystal (1.1 g, yield = 93%). The NCA was stored at −20 °C for 2 months.

**¹H NMR** (400 MHz, DMSO-$d_6$) δ 8.97 (d, J = 3.6 Hz, 1H), 4.32 (dt, J = 7.2, 3.6 Hz, 1H), 3.70 (s, 3H), 3.23 (dd, J = 16.7, 7.2 Hz, 1H), 2.90 (dd, J = 16.7, 3.6 Hz, 1H).

**¹³C NMR** (101 MHz, DMSO-$d_6$) δ 170.9, 165.1, 148.8, 52.9, 48.1, 30.6.

**HRMS** (EI, m/z): $[M + H]^+$ calculated for $C_6H_8NO_5^+$: 174.0397; found: 174.0396.

**FT-IR** (cm⁻¹) 1807, 1766.

## Data availability

The authors declare that all data supporting the findings of this study are available within the paper and its supplementary information files. The accession numbers for the crystallographic data reported in this paper are CCDC: 2052323, 2052324, 2052321, and 2088683. These data can be obtained free of charge from the Cambridge Crystallographic Data Centre at https://www.ccdc.cam.ac.uk/structures. Extra data are available from the corresponding author upon request.

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

## Acknowledgements

This work is supported by the National Natural Science Foundation of China (The National Science Fund for Distinguished Young Scholars No. 22125101 and General Project No. 21975004) and Li Ge-Zhao Ning Life Science Research Foundation for Young Scholars (LGZNQN202006). Z.-Y. T. was supported by the National Postdoctoral Program for Innovative Talents of China (BX20190004). The computation was supported by the High-performance Computing Platform of Peking University. The authors thank Dr. Feng Pan for the X-ray diffraction of crystals. We would also like to thank Jialing Sun, Dr. Yali Hu, Dr. Guangqi Wu for their help on polymerization data and Dr. Wei Xiong for the MALDI-TOF mass spectrum.

## Author contributions

Z.-Y.T. and H.L. conceived the idea, designed the experiments, analyzed the data, and wrote the manuscript. Z.-Y.T. conducted most experiments and the DFT calculation; Z.Z. conducted and analyzed the copolymerization of EG₃GluNCA and CysNCA. S.W. reproduced key experiments and helped the recrystallization of CysNCA. All of the authors read and approved the final version of the manuscript.

## Competing interests

The authors declare no competing interests.
