## [Peer Review File · Nature Communications]

Reviewers' Comments:

Reviewer #1:

Remarks to the Author:

This manuscript reports an original approach towards N-carboxyanhydrides and O-carboxyanhydrides monomers preparation. Briefly, this work shows that using epoxy compounds during syntheses based on the Fuchs-Farting method (and not the Leuch method) is a way 1) to optimize the preparation of known monomers and 2) to afford the preparation of new monomers with unprotected side chains (and this is remarkable). Overall, the work is comprehensively described and the manuscript may represent a significant step further for the scientific community in the preparation and application of synthetic polypeptides polymers. The broad scope and the outstanding functional group tolerance of the proposed methodology is demonstrated. Nevertheless and in contrast, enhanced access to polypeptides in "one step synthesis" is not sufficiently substantiated but is an important point to validate the proposed approach. This manuscript contains some interesting observations, but more substantial experimental work regarding ROP is necessary to demonstrate how, or until where, the proposed new monomers are superior in practice to the already used NCA (see my comments below). Hence, I could only recommend the publication of this manuscript in Nature Communication if my comments are taken into account.

I have four main comments regarding the work submitted by Hua Lu and coworkers:

1) As indicated by the authors, a major difficulty in the synthesis of NCA/OCA monomers comes from their purification. While the authors present their methodology as superior to previous methodologies, the manuscript does not provide any comparative data corresponding to storage (compared to conventional syntheses) or polymerizability. First, and concerning NCA storage, a comparison of the monomer degradation in anhydrous solvent and at high temperature would be a way to obtain data quickly (pinene versus epoxy compounds for instance, NMR or FTIR analyses). Second, standard polymerization kinetics (e.g. hexylamine in DMF, M/I = 100) would allow a good evaluation of the NCA obtained using alpha pinene versus the proposed method. Using some known monomers (BLG-NCA etc... see scheme 2), these data are important to provide. As a minor comment, the by-products resulting from the opening of epoxy compounds are not as volatile as the authors claim. I find a boiling point of 125 or 133°C depending on the epoxy compound used.

2) The mechanistic study proposed by the authors is not really convincing. For example, Figure 1B: why not comparing with similar stoichiometry? What about the same reaction performed in DCM? Figure S1 and S2: The CH signals that appear using DCI seem to reveal the presence of oligomers or other secondary products that are not discussed. Figure 1C and molecular modeling: the diagram is confusing, TS3a and TS3c should be aligned as well as TS4a and TS4c or Int5a or Int5c. The energy differences of the transition states are too small to explain the difference observed experimentally in Figure 1B. What about the transition states calculated by replacing PO with pinene (or just ethylene)?

3) I think that the discussions around the monomers presented in scheme 2 are too long and that the main message of the article becomes weak or confusing (BLG-NCA, Glycine-NCA etc...). It is necessary to put more emphasis on the monomers presented in scheme 3. This comment also applies to the abstract and the introduction. Considering the state of the art, the tolerance of a synthesis towards traces of water is much less original than being able to access these new NCA monomers. This is the real interest of the proposed article (the generation and the polymerization of challenging NCA thanks to the proposed methodology).

4) In line with the previous comment, it would be important to further develop the polymerization of GluNCA or CysNCA. Authors should give some examples with different degrees of polymerization, they may also indicate possible solubility issues and discuss the influence of acidic side chains during the propagation involving amines (I am skeptical about this last reactivity which takes place at room temperature; I would have expected a polymerization but at a higher temperature). Overall, each ROP example must be perfectly characterized (NMR, SEC) and provided in an extra table. Moreover, in the case of Glu-NCA, a MALDI analysis would eliminate any doubt about the nature of the polypeptide that is formed. Concerning the synthesis of these

challenging monomers, it is surprising that secondary reactions with epoxides or phosgene are not observed. How do the authors explain this?

5) Minor comments

a) Reactions with triphosgene are usually heated to decompose the triphosgene, not to make the NCA ring. The remark is similar with diphosgene and activated charcoal (OCA synthesis). In both cases, the direct use of phosgene allows synthesis without heating and without activated charcoal. The corresponding discussions should be revised.

b) I do not understand the one-step synthesis presented by the authors. There is indeed a synthesis step, followed by purification and then a polymerization. There are at least several steps for all the monomers presented.

c) For solubility reasons, I understand the need to copolymerize cysteine. On the other hand, I am skeptical about side-chain initiation from thiol functions. The literature describes very well that this type of initiator significantly increases polydispersity. On the other hand, the disappearance of thiol functions can also mean that disulfide bridges have been generated. The authors must give more characterizations (NMR analyses for example) to support their assertion.

Reviewer #2:

Remarks to the Author:

The manuscript titled " A Robust, Moisture-Tolerant, and Scalable Route to Unprotected α/β -Amino Acid N-Carboxyanhydrides and One-Step Synthesis of Hyperbranched Polypeptides " written by Z. Tian and H. Lu reported practical and scalable synthesis of NCAs containing a variety of functional groups and their use for ROP. As the authors described, NCAs are really important for preparing polypeptides. The development of high-yielding, scalable, and facile synthetic approaches for structurally diverse NCAs remains a highly important pursuit. The authors used epoxides to rapidly remove in situ generated HCl that causes undesired decomposition of NCAs. This is a great idea. The used epoxides are inexpensive and commercially available. They rapidly trapped both Cl ion and proton to prevent the side reactions at ambient temperature. In addition, the resultant chlorohydrins can be removed very easily. Surprisingly, even NCAs containing highly nucleophilic alcohol or thiol moiety can be prepared. Strict dehydration conditions were not required. Although, this reviewer have experienced NCA synthesis for a long time, I never supposed that the NCAs are stable in the presence of nucleophilic water, alcohol and thiol. As the authors mentioned, the purification of NCAs is really tricky. In order to obtain highly pure NCAs, repeated recrystallizations are sometimes required. However, the developed synthetic approach enabled simple preparation and purification of NCAs that can be used for ROP. This is a significant achievement in NCA-related chemistry. Because many polymer chemists have troubles in preparing NCAs. All the compounds were adequately characterized by NMR, MS, IR, HRMS, and X-ray crystallographic analysis. Overall, I think this manuscript is suitable for publication in Nature Communications.

Please consider several points shown below before publication.

Page 2, line 6 from bottom

Leuch's method -> Leuchs's method

Page 5, line 9

L-analine -> L-alanine

Although I know NCAs are not highly racemizable, it should be better to check racemization during NCA formation. The most racemizable CysNCA formation should be checked.

Page 10, line 5

Fuse and coworkers reported not only ^1H NMR yields but isolated yields of β -NCAs in ref. 54 (Scheme 4).

Reviewer #3:

Remarks to the Author:

The authors describe the application of an acid scavenger epoxide to established NCA forming reactions to allow for ring closure under ambient, hydrous conditions and with free alcohol groups. A panel of different monomers were prepared. Overall, the motivation for the work lacks clarity. NCAs with alcohols, amines, acids etc typically have protecting groups that serve to prevent interference of these groups with polymerization initiators/catalysts, to improve organic solubility and ease of purification, and to avoid unwanted initiation by these groups so that the NCAs are stable for long-term storage.

The study does not address the important issues of storage stability of these unprotected NCAs, nor their purity for polymerizations. The authors suggest they have good polymer molecular weight control with only a single PLG sample, and the NMR of Glu NCA (Fig S31) does not appear pure as claimed. Further, proton NMRs can't confirm the concentration of HCl for example, and trace impurities can inhibit many common initiators. Without polymerization and storage data, it is impossible to tell whether or not addition of an epoxide improves product quality.

The standard purification techniques of recrystallizations, filtrations, aqueous workup, or flash chromatography are all still used so there is not a procedural improvement in this area. The NCA product yields are not improved over prior published procedures either. Many of the NCAs in scheme 2 are readily prepared by simple treatment with phosgene or triphosgene and no acid scavenger is needed (dozen of publications). The advantage of epoxides vs acid scavengers TEA or pinene is not clear. Removal of TEA salts by filtration is reported in many papers and seems trivial, as opposed to the insinuation by the authors it is challenging. Even if epoxides scavenge faster than pinene the advantage is not stark enough to warrant publication in Nat. Comm.

The advantage of the epoxide seems to be that it in the hour the NCA reaction is stirring in open air the NCA doesn't undergo major hydrolysis or polymerization. The ability to stir in open air vs with a septa and N₂ is not a strong advantage that warrants publication in Nat. Comm. It is interesting that they are able to synthesis a few canonical amino acids without protecting groups, but the motivation to have free alcohols on Ser/Thr for example is unclear. PolySer for example readily forms insoluble beta sheets and isn't a material in high demand. The ability of Cys NCA to initiate other NCAs begs the questions the stability and applicability of this NCA, does this only happen in basic or polar conditions?

Additionally, the figures are of poor quality to the point it is difficult to read them and Fig 1B is particularly obscured/grainy. There is a typo in line 126 and • Scheme 1 says N₂ protection which reads as protection of dinitrogen

While the applicability of this paper with NCAs is minimal, it might have a bigger impact with OCAs and beta amino acids. Overall this work is not suitable for Nat. Comm., but could be improved into a methods paper or communication in a more specialized journal.

Additional experiments and major changes include:

1. Ring-opening Polymerization (ROP) results of BnGluNCA, EG₃GluNCA, and Sar NCA (Table S2 and Figure S25).
2. More ROP attempts of GluNCA, copolymerization of CysNCA, and more thorough characterizations (SEC, MALDI-TOF, NMR...etc) of the homo- and co-polymers (Figure 2 and S32-S39).
3. Comparison of the HCl-quenching efficiency of PO and pinene in DCM.
4. Tris(2-carboxyethyl)phosphine (TCEP) reduction of the hyperbranched polymer P(EG₃-Glu)-*branch*-PCys, which ruled out the hypothesis of potential formation of disulfide crosslinking (Figure S35).
5. Description regarding the storage conditions of various NCAs (See SI).
6. More discussion on Figure 2 and reduced length of narrative contents on Scheme 2.
7. Figures and Schemes are redrawn.
8. Language polishing, corrections of grammar and typos, and other minor changes in accordance with reviewers' request.

Point-to-point response.

Reviewer #1:

This manuscript reports an original approach towards N-carboxyanhydrides and O-carboxyanhydrides monomers preparation. Briefly, this work shows that using epoxy compounds during syntheses based on the Fuchs-Farting method (and not the Leuch method) is a way 1) to optimize the preparation of known monomers and 2) to afford the preparation of new monomers with unprotected side chains (and this is remarkable). Overall, the work is comprehensively described and the manuscript may represent a significant step further for the scientific community in the preparation and application of synthetic polypeptides polymers. The broad scope and the outstanding functional group tolerance of the proposed methodology is demonstrated. Nevertheless and in contrast, enhanced access to polypeptides in "one step synthesis" is not sufficiently substantiated but is an important point to validate the proposed approach. This manuscript contains some interesting observations, but more substantial experimental work regarding ROP is necessary to demonstrate how, or until where, the proposed new monomers are superior in practice to the already used NCA (see my comments below). Hence, I could only recommend the publication of this manuscript in Nature Communication if my comments are taken into account.

Our response: We sincerely welcome and appreciate these comments from the reviewer 1.

I have four main comments regarding the work submitted by Hua Lu and coworkers:

1) As indicated by the authors, a major difficulty in the synthesis of NCA/OCA monomers comes from their purification. While the authors present their methodology as superior to previous methodologies, the manuscript does not provide any comparative data corresponding to storage (compared to conventional syntheses) or polymerizability. First, and concerning NCA storage, a comparison of the monomer degradation in anhydrous solvent and at high temperature would be a way to obtain data quickly (pinene versus epoxy compounds for instance, NMR or FTIR analyses). Second, standard polymerization kinetics (e.g. hexylamine in DMF, M/I = 100) would allow a good evaluation of the NCA obtained using alpha pinene versus the proposed method. Using some known monomers (BLG-NCA etc... see scheme 2), these data are important to provide.

Our response:

For long-term storage: NCAs are known to be sensitive to moisture, so they are normally stored in a glovebox at low temperature. In addition, the shelf-life of monomers in bulk is highly dependent on the side groups. We stored most of the NCAs prepared in this manuscript under ambient atmosphere without desiccators, but low temperatures were still indispensable. For example, hydrophobic NCAs such as Bn-GluNCA could remain stable at -10 °C for at least 5 months, whereas the hydroxyl-bearing SerNCA and ThrNCA were stable for 1 month under the same condition. Both CysNCA and GluNCA can be stored at -20 °C for at least 1 month. Nevertheless, the hydroscopic EG₃GluNCA needs to be stored in a freezer at -30 °C inside a glovebox, in consistence with what previous studies described. We have included the information in the revised SI (at the end of the synthetic procedures for each monomer).

For polymerizability: we tested the ROP of Bn-Glu NCA, Sar NCA, and EG₃-GluNCA (Table S2 and Figure S25), three most frequently used monomers in our lab. Our results indicated that each of these monomers produced the corresponding polypeptide with a well-defined structure, a predictable molar mass (M_n), and narrow dispersity (\mathcal{D}). For example, when conducted in parallel, benzyl amine (BnNH₂)-mediated ROP of BLG-NCA prepared from the traditional (entry 1-2) method showed poorer M_n control compared to the same reaction using monomers that were synthesized by our new method (entry 3-5). Moreover, the good reproducibility was demonstrated by our observation that the M_n and \mathcal{D} of the polypeptide products remained largely unchanged regardless of the monomer batch or who performed the synthesis (entry 6&7, entry 8-11, and entry 17-20). We also tested a variety of initiators, including BnNH₂ (entry 3-5), hexyl amine (entry 14 and 16), butyl amine (entry 13), cholesterol amine (entry 15), a four-armed amine initiator (4A-NH₂, entry 17-20), Hexamethyldisilazane (HMDS, entry 8-12), and trimethylsilyl phenylsulfide (PhSTMS, entry 6-7). The results revealed that our new method could confer satisfactory and consistent control of both M_n and \mathcal{D} . Based on our experience, PhSTMS-mediated ROP is very sensitive to the purity of monomer. All together, we hope that

these experimental data provide sufficient evidence that the NCAs prepared by the method detailed in this study (with PO or ECH) have the same, if not higher, quality and purity as those afforded by the traditional approach.

Entry	Monomer	Initiator	[M] ₀	[M] ₀ /[I] ₀	Time	Conver.	Solvent	MW _{cal} (×10 ⁴ g mol ⁻¹)	MW _{obt} (×10 ⁴ g mol ⁻¹)	Đ
1	Bn-Glu NCA (Tradition)	BnNH ₂	0.20 M	50	12	> 99%	DMF	1.1	0.7	1.24
2	Bn-Glu NCA (Tradition)	BnNH ₂	0.20 M	100	12 h	> 99%	DMF	2.2	1.5	1.05
3	Bn-Glu NCA	BnNH ₂	0.20 M	50	12 h	> 99%	DMF	1.1	1.5	1.05
4	Bn-Glu NCA	BnNH ₂	0.20 M	100	12 h	> 99%	DMF	2.2	2.2	1.05
5	BLG/BDG	BnNH ₂	0.20 M	50	22 h	> 99%	DMF	1.1	1.0	1.06
6	EG ₃ -GluNCA (Batch 1)	PhS-TMS	0.15 M	75	24 h	> 99%	DMF	2.1	2.0	1.07
7	EG ₃ -GluNCA (Batch 2)	PhS-TMS	0.15 M	75	20 h	> 99%	DMF	2.1	2.0	1.06
8	EG ₃ -GluNCA (Batch 2)	HMDS	0.15 M	73	24 h	> 99%	DMF	2.0	2.0	1.13
9	EG ₃ -GluNCA (Batch 3)	HMDS	0.31 M	73	12 h	> 99%	DMF	2.0	1.9	1.05

10	EG ₃ -GluNCA (Batch 3)	HMDS	0.31 M	73	18 h	> 99%	DMF	2.0	1.9	1.07
11	EG ₃ -GluNCA (Batch 3)	HMDS	0.31 M	73	12 h	> 99%	DMF	2.0	1.9	1.07
12	EG ₃ -GluNCA (Batch 4)	HMDS	0.31 M	77	22 h	> 99%	DMF	2.1	1.9	1.08
13	EG ₃ -GluNCA (Batch 2)	Butylamine	0.31 M	75	12 h	> 99%	DMF	2.1	1.9	1.13
14	EG ₃ -GluNCA (Batch 5)	Hexylamine	0.31 M	50	12 h	> 99%	DMF	1.4	1.2	1.16
15	EG ₃ -GluNCA	Cholesterol- NH ₂	0.2 M	40	22 h	> 99%	DMF	1.1	1.1	1.06
16	SarNCA	Hexylamine	1.1 M	150	12 h	> 99%	DMF	1.1	1.0	1.07
17	SarNCA Batch 1	4A-NH ₂	1.0 M	250	16 h	> 99%	DMF	1.8	1.8	1.01
18	SarNCA (Batch 2)	4A-NH ₂	1.0 M	250	16 h	> 99%	DMF	1.8	1.7	1.04
19	SarNCA (Batch 3)	4A-NH ₂	1.0 M	250	16 h	> 99%	DMF	1.8	1.7	1.01
20	SarNCA (Batch 4)	4A-NH ₂	1.0 M	250	16 h	> 99%	DMF	1.8	1.8	1.01

We respectfully disagree with the reviewer on the necessity of testing monomer stability in anhydrous solvents and at high temperatures, primarily because the suggested condition is not what has been typically used for NCA storage. We would like to ask the reviewer to kindly clarify why he/she thinks such studies are relevant before we could.

As a minor comment, the by-products resulting from the opening of epoxy compounds are not as volatile as the authors claim. I find a boiling point of 125 or 133 °C depending on the epoxy compound used.

Our response: We greatly appreciate the reviewer's correction and have revised the statement, which now reads as follow (changes are highlighted in yellow):

“PO, 1-chloro-2-propanol, and 2-chloro-1-propanol are easy to remove during the workup.”

2) The mechanistic study proposed by the authors is not really convincing. For example, Figure 1B: why not comparing with similar stoichiometry? What about the same reaction performed in DCM?

Our response: We apologize for the confusion caused by putting “5:1” beneath the arrow in the reaction scheme. To clarify, experimental comparison of PO and pinene in Figure 1B was performed at a one-to-one molar ratio. By 5:1, we meant that the molar ratio of the two products, 1-chloro-2-propanol and 2-chloro-1-propanol, was 5:1, which was measured by ¹H NMR (Figure S4). We have moved the “5:1” to the correct location to avoid confusion. We initially performed the study in THF because it is the most frequently used solvent for NCA preparation. DCM is rarely used for NCA synthesis due to its low boiling point and the fact that amino acids generally have low solubility in DCM. Nevertheless, we also conducted the same experiments in DCM and obtained similar results as those in THF. Specifically, with DCM as solvent, PO quenched HCl within 3 min, whereas the use of pinene resulted in less than 50% conversion even at 4 h (Shown below).

entry	time (min)	conversion (%)
1	0	0
2	3	100

Overlay of ^1H NMR spectra of the (PO + DCI) reaction mixture in $\text{DCM-}d_2$ at time 0 and 3 min.

entry	time (h)	conversion (%)
1	0	0
2	1	43
3	2	44
4	4	47

Overlay of ^1H NMR spectra of the (pinene + DCl) reaction mixture in $\text{DCM-}d_2$ at different time points.

Figure S1 and S2: The CH signals that appear using DCI seem to reveal the presence of oligomers or other secondary products that are not discussed.

Our response: We agree with the reviewer that the peak broadening in Figure S1 (without DCI) indicated polymer generation, which had been mentioned in the manuscript. For Figure S2, the white precipitate in the NMR tube (inserted photo) indicated that most of the NCAs were hydrolyzed into free amino acids. This hypothesis was now confirmed by redissolving the precipitate in D₂O and analyzed by ¹H NMR (shown below). The broadening of the peak in the original Figure S2 was due to amino acid precipitation, which makes the shimming extremely difficult and unsatisfactory. This discussion was available on page S23 of SI. **We have added the new ¹H NMR spectrum of the precipitate as Figure S2B in the revised SI.**

Figure 1C and molecular modeling: the diagram is confusing, TS3a and TS3c should be aligned as well as TS4a and TS4c or Int5a or Int5c. The energy differences of the transition states are too small to explain the difference observed experimentally in Figure 1B. What about the transition states calculated by replacing PO with pinene (or just ethylene)?

Our response: We greatly appreciate the reviewer's suggestion and have rearranged Figure 1C. The rate-determining step of standard phosgenation is the one from Int4a to TS3a, which has an activation Gibbs free energy of a 23.4 kcal/mol. In comparison, the same step in the epoxide-assisted pathway has an activation Gibbs free energy of 19.2 kcal/mol. Based on the Curtin-Hammett principle and Eyring equation, the 4.2 kcal/mol gap between TS3a and TS3c could translate into a 1200-fold difference in reaction rate. We regret that in our first draft we made a typo there by incorrectly calculating the energy barrier difference to be 2.9 kcal/mol. The diagram in Figure 1C was modified as follow:

3) I think that the discussions around the monomers presented in scheme 2 are too long and that the main message of the article becomes weak or confusing (BLG-NCA, Glycine-NCA etc...). It is necessary to put more emphasis on the monomers presented in scheme 3. This comment also applies to the abstract and the introduction. Considering the state of the art, the tolerance of a synthesis towards traces of water is much less original than being able to access these new NCA monomers. This is the real interest of the proposed article (the generation and the polymerization of challenging NCA thanks to the proposed methodology).

Our response: We have rewritten the said paragraph in a more concise manner as suggested. Moreover, we added more discussion on the new NCA monomers.

4) In line with the previous comment, it would be important to further develop the polymerization of GluNCA or CysNCA. Authors should give some examples with different degrees of polymerization, they may also indicate possible solubility issues and discuss the influence of acidic side chains during the propagation involving amines (I am skeptical about this last reactivity which takes place at room temperature; I would have expected a polymerization but at a higher temperature). Overall, each ROP example must be perfectly characterized (NMR, SEC) and provided in an extra table. Moreover, in the case of Glu-NCA, a MALDI analysis would eliminate any doubt about the nature of the polypeptide that is formed. Concerning the synthesis of these challenging monomers, it is surprising that secondary reactions with epoxides or phosgene are not observed. How do the authors explain this?

Our response: We thank the reviewer for the suggestion. We have performed additional ROPs of GluNCA with varying feeding M/I ratios. The SEC traces of the ROP mixture (taken in situ, Figure 2B) confirmed the controlled ROP of GluNCA. The ¹H NMR (Figure S33) and MALDI-TOF (Figure S32) spectra proved that the polymer consisted of repeating units of 129 Da (the mass of Glu), with C₆H₅NH—/—H and end groups.

Figure S1 MALDI-TOF mass spectrometry of poly(L-glutamic acid).

We were also surprised to not have observed any solubility issue of the PLG in DMF even at a pretty high monomer concentration (500 mg/mL). However, once precipitated, the lyophilized PLG powder cannot be redissolved in DMF and was only soluble in alkaline aqueous buffers. We also did not observe a strong, detrimental effect of the side chain -COOH on the rate of the ROP. In fact, GluNCA was almost completely converted (>95%) within 2-3 h, with purification yields in the range of 75-93%. In an additional control study, we found that the addition of benzoic acid (equal molar to NCA) did not alter the rate of ROP of Z-LysNCA or Bn-GluNCA significantly (data not shown). This unexpectedly high polymerizability warrants further mechanistic investigation.

Despite being theoretically possible from a thermodynamic perspective, the side reactions of -CO₂H/-SH with epoxide/phosgene were not observed in our study, probably because they were too slow under our experimental conditions (r.t. or below) and without a proper acid/base catalyst (note that the HCl was immediately quenched by PO).

5) minor points

a) Reactions with triphosgene are usually heated to decompose the triphosgene, not to make the NCA ring. The remark is similar with diphosgene and activated charcoal (OCA synthesis). In both cases, the direct use of phosgene allows synthesis without heating and without activated charcoal. The corresponding discussions should be revised.

Our response: We understand the reviewer's concern but would like to point out that triphosgene has been reported to be thermally stable and only show a slight degree of decomposition until reaching its boiling point around 206 °C. However, triphosgene can be nucleophilically activated at room temperature (*Angew. Chem., Int. Ed.* **1987**, *26*, 894; *Org. Process Res. Dev.* **2017**, *21*, 1439).

b) I do not understand the one-step synthesis presented by the authors. There is indeed a synthesis step, followed by purification and then a polymerization. There are at least several steps for all the monomers presented.

Our response: We apologize for our choice of word and would like to clarify that we were referring to the fact that we could produce the hyperbranched polymer without the protection and deprotection of the side chains, unlike the previously reported multi-step synthesis (ref. 51). Nevertheless, we have replaced "one-step" with "facile" in the revised manuscript.

c) For solubility reasons, I understand the need to copolymerize cysteine. On the other hand, I am skeptical about side-chain initiation from thiol functions. The literature describes very well that this type of initiator significantly increases polydispersity. On the other hand, the disappearance of thiol functions can also mean that disulfide bridges have been generated. The authors must give more characterizations (NMR analyses for example) to support their assertion.

Our response:

We believe that the formation of disulfide is very unlikely because it usually requires strongly alkaline conditions with a high concentration of oxygen or some other oxidant (Nicol, E.; Bonnans-Plaisance, C.; Levesque, G. *Macromolecules* **1999**, *32*, 4485.). To rule out this possibility, we have treated the copolymer with TCEP, a well-known S-S reducing agent, and observed no shift of the SEC peak (Figure S35).

Figure S2. Overlay of the aqueous SEC traces of P(EG₃-Glu)-branch-Pcys before (black) and after (red) treatment of TCEP.

Our previous study indicated that thiol or related species could be used as initiator for the ROP of many different NCAs, affording polypeptides bearing thioester functionalities at the chain end (*Biomacromolecules* **2016**, *17* (3), 891-896; *J. Am. Chem. Soc.* **2016**, *138* (34), 10995-11000; *ACS Macro Lett.* **2018**, *7* (8), 892-897.). Therefore, it is not surprising that CysNCA can be used as an inimer for synthesizing the hyperbranched polymer in the current study. The increased dispersity of our product ($D = 1.66$, Figure 2E), as suggested by the reviewer, actually confirmed the multiple initiation pathway. To

add more evidence, we performed the copolymerization of CysNCA with BnGluNCA. The characterization of the copolymers were available in Figure S34-39, including ^1H NMR (Figure S34 and S37), H-H COSY (Figure S38), H-C HSQC (Figure S39). The existence of thioesters were also confirmed by the SEC shift after the copolymers were treated with cysteine in water (Figure 2E) or BnSH/TEA in DMF (Figure S36).

Figure S3 H-H COSEY spectra of PBLG-branch-Pcys. The resonances in the red circle were assigned to the correlation of α -H and β -H of Cys.

Figure S4 H-C HSQC spectra of PBLG-branch-Pcys. The resonances in the red circle were assigned to the correlation of α -H with α -C of Cys.

Reviewer #2:

The manuscript titled " A Robust, Moisture-Tolerant, and Scalable Route to Unprotected α/β -Amino Acid N-Carboxyanhydrides and One-Step Synthesis of Hyperbranched Polypeptides " written by Z. Tian and H. Lu reported practical and scalable synthesis of NCAs containing a variety of functional groups and their use for ROP. As the authors described, NCAs are really important for preparing polypeptides. The development of high-yielding, scalable, and facile synthetic approaches for structurally diverse NCAs remains a highly important pursuit. The authors used epoxides to rapidly remove in situ generated HCl that causes undesired decomposition of NCAs. This is a great idea. The used epoxides are inexpensive and commercially available. They rapidly trapped both Cl ion and proton to prevent the side reactions at ambient temperature. In addition, the

resultant chlorohydrins can be removed very easily. Surprisingly, even NCAs containing highly nucleophilic alcohol or thiol moiety can be prepared. Strict dehydration conditions were not required. Although, this reviewer have experienced NCA synthesis for a long time, I never supposed that the NCAs are stable in the presence of nucleophilic water, alcohol and thiol. As the authors mentioned, the purification of NCAs is really tricky. In order to obtain highly pure NCAs, repeated recrystallizations are sometimes required. However, the developed synthetic approach enabled simple preparation and purification of NCAs that can be used for ROP. This is a significant achievement in NCA-related chemistry. Because many polymer chemists have troubles in preparing NCAs. All the compounds were adequately characterized by NMR, MS, IR, HRMS, and X-ray crystallographic analysis. Overall, I think this manuscript is suitable for publication in Nature Communications.

Our response: We sincerely welcome and appreciate these comments from the reviewer 2.

Page 2, line 6 from bottom
Leuch's method -> Leuchs's method

Page 5, line 9
L-analine -> L-alanine

Our response: We have corrected the word in the revised manuscript.

Although I know NCAs are not highly racemizable, it should be better to check racemization during NCA formation. The most racemizable CysNCA formation should be checked.

Our response: We gratefully appreciate the reviewer's suggestion. Racemization during NCA synthesis has not been reported to the best of my knowledge, and we anticipate no racemization in our current study because our reaction conditions were milder than those employed previously. The absence of racemization in this study is also evidenced by:

1. While ^1H NMR cannot tell whether there is racemization or not for most NCAs, the racemization of the α -carbon of ThrNCA would generate diastereomers that

are distinguishable by ^1H NMR. Therefore, the lack of such ^1H NMR signals in our studies argues against racemization.

2. Some of the polymers generated from our NCAs showed high helicity (e.g. Figure 2D), which indicated the preservation of their chirality.
3. The X-ray crystal structure of monomers also showed very little racemization.

Fuse and coworkers reported not only ^1H NMR yields but isolated yields of β -NCAs in ref. 54 (Scheme 4).

Our response: We have deleted the inaccurate statement.

Reviewer #3:

The authors describe the application of an acid scavenger epoxide to established NCA forming reactions to allow for ring closure under ambient, hydrous conditions and with free alcohol groups. A panel of different monomers were prepared. Overall, the motivation for the work lacks clarity. NCAs with alcohols, amines, acids etc typically have protecting groups that serve to prevent interference of these groups with polymerization initiators/catalysts, to improve organic solubility and ease of purification, and to avoid unwanted initiation by these groups so that the NCAs are stable for long-term storage.

Our response: We agree with the reviewer that side chain protection has its benefits in many scenarios. However, the protection and deprotection steps are laborious, time-consuming, inefficient, and often requires costly and/or harsh conditions (e.g. the deprotection of PBLG involves 33% HBr/TFA, which can cause racemization if not careful). Therefore, the main purpose of our current study is to develop a method capable of synthesizing important and desirable polypeptide materials in a more rapid, efficient, and economical manner. We have demonstrated the utility of two of these unprotected NCAs (Figure 2). The first is the direct preparation of PLG from GluNCA, which shows good reaction control and higher yield, while using fewer steps, less time, and no harsh conditions (new data available in Figure 2; Table S2 and S25, and Figure S32-39). This is certainly beneficial and important for industrial, particularly biomedical, applications. The second example is the facile synthesis of cysteine-degradable hyperbranched polypeptides, which could be employed in a variety of biological

applications, including drug delivery, protein modification, surface and nanoparticle coating...etc.

The study does not address the important issues of storage stability of these unprotected NCAs, nor their purity for polymerizations. The authors suggest they have good polymer molecular weight control with only a single PLG sample, and the NMR of Glu NCA (Fig S31) does not appear pure as claimed. Further, proton NMRs can't confirm the concentration of HCl for example, and trace impurities can inhibit many common initiators. Without polymerization and storage data, it is impossible to tell whether or not addition of an epoxide improves product quality.

Our response: We thank the reviewer for this important remark, which is identical to the first major comment of Reviewer 1. For regular (previously reported) NCAs, As we have demonstrated on page R2-R3 in this letter, both the NCA storage conditions and polymerization results (Table S2 and Figure S25) were almost identical to those reported previously. For the new NCAs, apart from ^1H and ^{13}C NMR, high resolution mass spectrometry, FT-IR, and XRD that can demonstrate their high purity, we also studied and characterized their ROPs as shown in Figure 2 and S32-39. Taken together, we hope the reviewer could now be convinced that the NCAs prepared by our new methods are indeed highly pure.

The standard purification techniques of recrystallizations, filtrations, aqueous workup, or flash chromatography are all still used so there is not a procedural improvement in this area. The NCA product yields are not improved over prior published procedures either. Many of the NCAs in scheme 2 are readily prepared by simple treatment with phosgene or triphosgene and no acid scavenger is needed (dozen of publications). The advantage of epoxides vs acid scavengers TEA or pinene is not clear. Removal of TEA salts by filtration is reported in many papers and seems trivial, as opposed to the insinuation by the authors it is challenging. Even if epoxides scavenge faster than pinene the advantage is not stark enough to warrant publication in Nat. Comm. The advantage of the epoxide seems to be that it in the hour the NCA reaction is stirring in open air the NCA doesn't undergo major hydrolysis or polymerization. The ability to stir in open air vs with a septa and N_2 is not a strong advantage that warrants publication in Nat. Comm.

Our response:

We respectfully disagree with the reviewer on these comments. We believe that this paper warrants publication in Nat. Commun. not just because the method that we developed shows robustness under moist conditions, but also because of the following advantages it confers (we would like to point out that these advantages were also recognized by both Reviewer 1 and 2 in general). First, our work provided new mechanistic insight into NCA synthesis and explained why previous acid scavengers had limited success (Figure 1 and Figure S3-4). Of note, the detrimental role of HCl in the formation of NCA has not been fully elucidated from a mechanistic perspective until this work. Second, the method detailed in this study allows synthesis of unprotected and challenging NCAs was made possible under mild conditions (Scheme 2). Third, the work also allows rapid access to important linear and novel hyperbranched polypeptides with higher yield, fewer steps, and more environmentally friendly procedures (Figure 2).

A side-by-side comparison between epoxy and α -pinene was presented in both Figure 1 and Scheme 3. In short, major advantages of PO/ECH over α -pinene include the striking increase in reaction rate, often by several orders of magnitude, the ability to instantly remove HCl under mild conditions, as well as the general ease of product purification. As such, PO/ECH enables the synthesis of challenging NCAs, for which α -pinene failed to afford in satisfactory purity under the same conditions.

The greatest limitation of TEA is not the removal of TEA·HCl salt, but its basicity. As we discussed in the manuscript, i) TEA can prematurely initiate the ROP of NCA when in excess, ii) TEA cannot quench the nucleophilic chloride, which is known to attack the carbonyls of NCAs to form isocyanates and affect the subsequent ROP. iii) synthesis of

unprotected SerNCA, GluNCA, and CysNCA bearing –OH, –COOH, and/or –SH with TEA as HCl scavenger is unfeasible due to the deprotonation of the side groups.

Our new method has clear “procedural improvement”. First, our method completely obviates the need for stringent anhydrous conditions, which are considered obligatory by previous methods, regardless of whether an acid scavenger is used. It is worth emphasizing that the presence of moisture is significantly more detrimental to the stability of hydrophilic NCAs than that of hydrophobic NCAs. For instance, our lab has been working on the hydroscopic EG₃GluNCA for over 7 years and we have never succeeded when the environmental humidity exceeds 40%, even with the most stringent air-free conditions. In contrast, the new protocol allowed us to easily obtain grams of pure EG₃GluNCA at ~70% humidity without solvent drying. Second, heating is often required by the previous methods, especially for amino acids with low solubility in THF. The ROPs that we developed in this study, on the other hand, can be conducted at relatively mild temperature, which is particularly crucial for synthesizing labile NCAs. Third, the workup procedures of our current method are much easier to execute and less demanding. For example, BnGluNCA and Z-LysNCA were previously recrystallized in a glove box, but this can now be performed in open air. Similarly, column chromatographic purification of NCAs can be conducted at practically any level of humidity, without the use of a glove box or dry solvent. Fourth, there is also clear evidence that our new method can lead to yield improvement, shorter reaction time, and simpler procedures. Taking the synthesis of ProNCA as a vivid example, we are presenting a side-by-side comparison between a literature protocol that we consider state-of-the-art (10.1021/bm200495n, left panel) and the method that we developed in this study (right panel) for the reviewer’s convenience.

Synthesis of L-Proline N-Carboxy Anhydride (LP-NCA). *tert*-Butyloxycarbonyl-L-proline (Boc-L-proline) (30 g, 0.139 mol) was dissolved in freshly distilled THF (600 mL) in a 1 L two-necked round-bottomed flask with a slight flow of dry Ar gas. Then, triphosgene (0.37 equiv, 15.2 g) was added under vigorous stirring. After 10 min, freshly distilled triethylamine (1.1 equiv, 21.3 mL) was added dropwise at 0 °C, leading to the formation of N-carboxy anhydride in a one-pot procedure with instantaneous precipitation of TEA·HCl salt. After stirring for 6.5 h at room temperature under Ar atmosphere, the reaction mixture was cooled at 0 °C to allow complete precipitation of the salt, and the precipitate was removed by filtration. The filtrate was then distilled under vacuum (to remove THF and excess phosgene), and the crude mixture was dried overnight in the high vacuum line.

Crude LP-NCA was then dissolved in ethyl acetate, chilled, and extracted²⁶ with ice-cold water (100 mL) until neutral pH. The organic phase was then separated, dried over MgSO₄, and filtered, and the solvent was distilled off in the vacuum line, yielding LP-NCA (8 g, yield ~50%).

Purification of LP-NCA. Crude LP-NCA was dissolved in the minimum amount of THF and a large excess of hexane (400 mL) was added, leading to a milky, thick precipitate which was cooled to -20 °C for complete precipitation. The solvent mixture was then removed under Ar filtration, using a microfilter candle for reversed filtration with a narrow tube (Duran, *d* = 13 mm, por.3), and the precipitate was dried on the high vacuum line. In this way, **1** (Scheme 1) was removed. The purification was monitored with FTIR and ¹H NMR spectroscopy.

The next step involved thermal dissolution of LP-NCA crystals in hexane. In this additional purification step, the dried crystals were suspended in a large amount of hexane (~ 500 mL) and gradually heated to 45 °C under vigorous stirring until the anhydride was thoroughly dissolved. The solution was then filtered immediately with a warm filter funnel (125 mL capacity, por.3) under vacuum in a new two-necked 1 L round-bottomed flask, whereas nonsoluble oily contaminants were removed. Very long needle-like white LP-NCA crystals were obtained when the flask was cooled to -20 °C. Cold hexane was filtered under Ar using a microfilter candle for reversed filtration, and the resulting LP-NCA crystals were dried under high vacuum. This procedure resulted to removal of **2** of Scheme 1. The procedure was repeated if contamination was indicated by FTIR spectroscopy. The purity of the final product was verified by FTIR and ¹H and ¹³C NMR spectroscopy, and electrospray ionization mass spectroscopy (ESI-MS), and it was stored in a glovebox. Overall yield: 6 g (30%) m.p.: 53–55 °C.

Overall, we feel the robustness, milder conditions, and time saving can all be considered procedure improvements. There is no doubt that they are crucial for reducing the cost, improving the reproducibility, and facilitating the scale up, of the synthesis of NCAs.

Our protocol of L-Proline NCA Synthesis

Air humidity: 74%

To a 100 mL flask, Boc-L-proline **2b** (1.0 g, 4.7 mmol, 1.0 eq), CH₃CN (15 mL), methyloxirane (3.7 mL, 47 mmol, 10.0 eq) were added sequentially under magnetic stirring. Triphosgene (700 mg, 2.3 mmol, 0.5 eq) was finally added in one portion and the adapter was slightly opening. The reaction was stirred at ~ 4 °C in an ice bath for 2.5 h. For safety reason, the excessive triphosgene was quenched by adding 0.5 mL cold water at ~ 4 °C with 1–3 min stirring. Then brine (1 mL) was added to facilitate phase separation and subsequent extraction. The aqueous phase was extracted with ethyl acetate (EA, 10 mL × 2). The combined organic phase was directly dried with anhydrous Na₂SO₄. After the removal of solvent by rotatory evaporation in vacuum under 40 °C, the crude product was purified by flash column chromatography (PE/EA = 5:1 ~ 1:1 ~ EA). The pure product **4b** was obtained as a light yellow oil (470 mg, yield = 72%).

It is interesting that they are able to synthesis a few canonical amino acids without protecting groups, but the motivation to have free alcohols on Ser/Thr for example is unclear. PolySer for example readily forms insoluble beta sheets and isn't a material in high demand. The ability of Cys NCA to initiate other NCAs begs the questions the stability and applicability of this NCA, does this only happen in basic or polar conditions?

Our response: While PolySer is insoluble and the homopolymerization of SerNCA could be problematic, Ser/Thr/Cys NCAs will be highly useful for copolymerization with other NCAs for synthesizing alcohol-containing polypeptides without protection/deprotection. CysNCA crystals can be stored in a -20 °C freezer, without a glovebox or desiccator, for at least 1 month. CysNCA dissolved in dry THF is stable for 1-2 h at room temperature. However, CysNCA in pure DMF solution would undergo rapid polymerization/decomposition triggered by trace amount of water at room temperature or higher. More rigorous investigation of the ROP of SerNCA, HypNCA, PenNCA, and CysNCA is currently ongoing in our lab and beyond the scope of this paper. We hope these results can be shared with colleagues in the field in the near future.

Additionally, the figures are of poor quality to the point it is difficult to read them and Fig 1B is particularly obscured/grainy. There is a typo in line 126 and • Scheme 1 says N2 protection which reads as protection of dinitrogen

Our response: We thank the reviewer for this important remark. We have polished the quality of ALL figure and thoroughly proofread our manuscript to remove such typos.

While the applicability of this paper with NCAs is minimal, it might have a bigger impact with OCAs and beta amino acids.

Our response: We agree with the Reviewer that this method could further make a significant impact on OCAs and β NCAs, which will be pursued in future studies.

Reviewers' Comments:

Reviewer #1:

Remarks to the Author:

This revised manuscript basically addressed all my previous concerns. All the questions have been brightly addressed in this revision. As ultimate minor correction:

The following sentence should be corrected: « The "Leuchs' method" for the generation of NCA, which involves the phosgenation and ring-closure of amino acids via (tri)phosgene, is usually carried out under strictly anhydrous and air-free conditions (Scheme 1). » The authors definitely mistake the Leuch method and the Fuchs-Farthing method. It is with the latter method that phosgenation is used. « Leuch » should therefore be replaced by « Fuchs-Farthing ».

I then recommend the manuscript for publication in Nature Communications and I greatly congrats all the authors for their contribution to the field and for their hard work.

Reviewer #2:

Remarks to the Author:

This reviewer 2 read point to point response and revised manuscript. Again, I suggest the authors to experimentally check the racemization of at least one highly racemizable NCA. NCAs are not highly racemizable, on the other hand, reportedly, N-protected NCAs are highly racemizable. Although I think the significant racemization did not occur under the developed mild reaction conditions, the experimental data to show the absence of racemization further increase the value of this study. The authors indicated no generation of diastereomer in the preparation of ThrNCA, however, Thr is not highly racemizable, therefore, the results from ThrNCA did not ensure that no racemization occurred in all the other NCAs. The most racemizable NCA in the submitted manuscript is CysNCA. X-ray crystallographic analysis of CysNCA is not reported in the submitted manuscript. Even < 1% of racemization is not negligible. I think the observed helicity (Figure 2D) did not ensure that such a tiny amount of racemization did not occur during the reaction. I was convinced about other points.

Reviewer #3:

Remarks to the Author:

The manuscript has been updated to address some, but not all issues brought forth during the previous review. As stated in the author response letter, the major innovation here is access to NCA monomers without protecting groups. Apparently, these do not cyclize without the addition of the epoxide compounds. The characterization of the branches Cys polymers is much improved. However, the utility and application of the PGA polymers still needs work to support the claims. In the new conclusion section, it is stated that they obtained "outstanding polymerization control" (there is actually a typo here), however, only 2 polymers were made at low M:I ratios. In order to make the claim of polymerization control, they should show polymers prepared at many M:I ratios and higher M:I ratios (i.e. 100, 200:1) and show chain extension experiments.

From the data in Figure 1, it's clear that water will initiate NCA polymerization in a matter of hours. While the epoxides clearly scavenge acid and accelerate cyclization into NCA compounds, it's not clear how the epoxide compounds prevent later NCA hydrolysis, polymerization, or improve storage conditions. After the cyclization reaction, the NCAs are subjected to aqueous workup, crystallization, or chromatography. The epoxide and related chloride byproducts should be removed at this point. The authors have failed to explain why addition of an acid scavenger that is removed during workup/purification would have any effect on long term monomer stability or result in a compound that is stable to recrystallization in "wet solvents". Why wouldn't the water initiate during recrystallizations just as it does in the Fig 1? NCAs still need to be stored and handled dry. Further, what is meant by "regular solvent" or "wet solvent"? this can vary wildly depending on brand, age, and storage conditions. The authors should use a titrator to determine the ppm water and to determine if this has any effect on the NCA cyclization, purity, and stability in their system. Until the authors can explain how the simple addition of an epoxide to the cyclization reaction results in resistance to hydrolysis after the epoxide is gone, I can't recommend

publication.

For the experiments comparing epoxide acid scavengers to pinene or "traditional" NCA synthesis, the experimental conditions are not well defined. How many equivs of pinene vs epoxide? What is "tradition" defined as? The reactions need to be compared with the same equivalents of pinene as epoxide. Most of the other monomers synthesized have been reported elsewhere. Some compounds have been reported in in hundreds of publications over many decades, and so speeding the reaction from 4 h to 1 h or dry vs wet solvent isn't viewed as a game changer.

While the substitution of an epoxide for pinene seems to be a nifty synthetic tip worthy of disseminating to those in the field, the innovation is not stark enough to warrant publication here. And if the true innovation is access to protecting group-free NCAs, this aspect should be expanded upon.

Additional experiments and major changes in the manuscript include:

1. Chain extension of GluNCA and EG₃GluNCA for making block polypeptides (Figure S34).
2. Acquiring the crystallographic structure of CysNCA (Figure S52)
3. BnGluNCA stability in regular THF (~ 2000 ppm water)
4. More discussion on the role of epoxy compounds on the workup and storage of NCAs.

Point-to-point response.

Reviewer #1 (Remarks to the Author):

This revised manuscript basically addressed all my previous concerns. All the questions have been brightly addressed in this revision. As ultimate minor correction: The following sentence should be corrected: « The “Leuchs’ method” for the generation of NCA, which involves the phosgenation and ring-closure of amino acids via (tri)phosgene, is usually carried out under strictly anhydrous and air-free conditions (Scheme 1). » The authors definitely mistake the Leuch method and the Fuchs-Farthing method. It is with the latter method that phosgenation is used. « Leuch » should therefore be replaced by « Fuchs-Farthing ». I then recommend the manuscript for publication in Nature Communications and I greatly congrats all the authors for their contribution to the field and for their hard work.

Our response: We are utmost thankful for the extremely helpful and constructive suggestions from Reviewer 1. The minor mistake has been corrected in the revised manuscript.

Reviewer #2 (Remarks to the Author):

This reviewer 2 read point to point response and revised manuscript. Again, I suggest the authors to experimentally check the racemization of at least one highly racemizable NCA. NCAs are not highly racemizable, on the other hand, reportedly, N-protected NCAs are highly racemizable. Although I think the significant racemization did not occur under the developed mild reaction conditions, the experimental data to show the absence of racemization further increase the value of this study. The authors indicated no generation of diastereomer in the preparation of ThrNCA, however, Thr is not highly racemizable, therefore, the results from ThrNCA did not ensure that no racemization occurred in all the other NCAs. The most racemizable NCA in the submitted manuscript is CysNCA. X-ray crystallographic analysis of CysNCA is not reported in the submitted manuscript. Even < 1% of racemization is not negligible. I think the observed helicity (Figure 2D) did not ensure that such a tiny amount of racemization did not occur during the reaction. I was convinced about other points.

Our response: We thanks Reviewer 2 for raising this important point. Per the request, the X-ray crystallographic analysis of CysNCA was acquired and the result was deposited in Cambridge Crystallographic Data Centre (CCDC-2088683). The report of CysNCA returns no Level_A or Level_B errors and shows less than 1% racemization (red box show below). Considering the starting material (L-cysteine from *Aladdin Inc.*) is also 99% pure, we think this result is enough to support our argument.

```
_refine_ls_abs_structure_details
```

```
;
Flack x determined using 582 quotients [(I+) - (I-)] / [(I+) + (I-)]
(Parsons, Flack and Wagner, Acta Cryst. B69 (2013) 249)
```

```
;
_refine_ls_abs_structure_Flack          -0.01 (3)
_refine_ls_extinction_coef              .
_refine_ls_extinction_method            none
_refine_ls_goodness_of_fit_ref          1.128
_refine_ls_hydrogen_treatment           constr
_refine_ls_matrix_type                   full
..
```

Reviewer #3 (Remarks to the Author):

The manuscript has been updated to address some, but not all issues brought forth during the previous review. As stated in the author response letter, the major innovation here is access to NCA monomers without protecting groups. Apparently, these do not cyclize without the addition of the epoxide compounds. The characterization of the branches Cys polymers is much improved. However, the utility and application of the PLG polymers still needs work to support the claims. In the new conclusion section, it is stated that they obtained “outstanding polymerization control” (there is actually a typo here), however, only 2 polymers were made at low M:I ratios. In order to make the claim of polymerization control, they should show polymers

prepared at many M:I ratios and higher M:I ratios (i.e. 100, 200:1) and show chain extension experiments.

Our response: We performed chain extension study from the consecutive ROP of GluNCA and EG₃GluNCA. The data were shown below. Both SEC (Figure 2C) and ¹H NMR (Figure 334) confirmed the successful generation of the block copolymer PLG-*block*-P(EG₃Glu).

Figure S34. ¹H NMR spectrum of PLG-*block*-P(EG₃-Glu) in D₂O (400 MHz).

Figure 2C. SEC of PLG and PLG-*block*-P(EG₃-Glu)

Prof. Kataoka have used PEG-PLG block copolymer as carriers to deliver various drugs, and those technologies have been pushed forward to nanomedicines in different stages of clinical trials (e.g. **NC-6004 and others, NanoCarrier Ltd.**). To further demonstrate the utility of

GluNCA, we also prepared PEG-*b*-PLG block copolymer by using PEGNH₂ (MW = 2000) as a macroinitiator.

Figure R1. SEC trace of PEG-*block*-PLG (MW of PEG = 2000, DP of PLG = 23)

Figure R2. ¹H NMR of PEG-*b*-PLG (MW=2000) in D₂O with NaOD

We believe that the current studies have sufficiently demonstrated its potential in facilitating the laboratory synthesis of PLG or PEG-PLG for its high step economy (from 4 steps to 2 steps), higher overall yield, and no need of using harsh deprotection condition (33% HBr-HOAc in TFA).

Its potential in innovating the industrial-scale preparation of biomedicine-grade and higher M_n PLG or PEG-PLG would be very important in the future, which is obviously out of the scope of this work.

From the data in Figure 1, it's clear that water will initiate NCA polymerization in a matter of hours. While the epoxides clearly scavenge acid and accelerate cyclization into NCA compounds, it's not clear how the epoxide compounds prevent later NCA hydrolysis, polymerization, or improve storage conditions. After the cyclization reaction, the NCAs are subjected to aqueous workup, crystallization, or chromatography. The epoxide and related chloride byproducts should be removed at this point. The authors have failed to explain why addition of an acid scavenger that is removed during workup/purification would have any effect on long term monomer stability or result in a compound that is stable to recrystallization in "wet solvents". Why wouldn't the water initiate during recrystallizations just as it does in the Fig 1? NCAs still need to be stored and handled dry. Further, what is meant by "regular solvent" or "wet solvent"? this can vary wildly depending on brand, age, and storage conditions. The authors should use a titrator to determine the ppm water and to determine if this has any effect on the NCA cyclization, purity, and stability in their system. Until the authors can explain how the simple addition of an epoxide to the cyclization reaction results in resistance to hydrolysis after the epoxide is gone, I can't recommend publication.

Our response: As we demonstrated in Figure 1A, NCA became sensitized to water when HCl presents. Thus, the NCA can easily go decomposition during workup for conventional methods as a large amount of HCl coexists. **The use of epoxy facilitated the preparation of NCAs under less stringent conditions by the means of removing HCl in a highly efficient, effective, and complete fashion, which desensitized the NCA to environmental water during cyclization, workup, and storage.**

We apologize for the confusing terms used in the manuscript and have rephrased those terms in the revised manuscript. For "wet solvents", we mean solvents with extra water added. For example, we wrote in our manuscript: "*The NCA synthesis succeeded even with intentionally added H₂O (1 eqiv to amino acid) to the reaction mixture.*" Here, the solvent is technically a "wet solvent". The same thing applies to the solvent used in Figure 1A (2M D₂O reached **180000 ppm by calculation**). For "regular solvents", we mean analytical grade solvents from commercial sources (see the table below).

entry	Content of water in ppm
THF- d ₈ used in Figure 1A (CIL Inc)	< 200 ppm
THF used for NCA preparation and recrystallization (analytical grade, Concord Technology Inc. (Tianjin, China))	< 2040 ppm
DMF used for PLG preparation (analytical grade, Concord	< 4000 ppm

Technology Inc. (Tianjin, China)	
ACN used for NCA preparation and recrystallization (analytical grade, Concord Technology Inc. (Tianjin, China))	< 1100 ppm
PE used for recrystallization and column chromatography (analytical grade, Tong Guang Chem. Inc., Beijing, China)	< 700 ppm
Ethyl acetate used for column chromatography (analytical grade, Tong Guang Chem. Inc., Beijing, China)	< 2300 ppm

The rate of NCA decomposition is highly related to water content. As seen from the above table, the solvents we used for NCA preparation, recrystallization, or column chromatography (regular solvents) contain much less amount of water than the “wet solvent” used in Figure 1A. The decomposition of BnGluNCA (0.2 mM) in regular THF (analytical grade, < 2040 ppm water) was not observed until 27 h (Figure R3).

Figure R3. Overlay of IR spectra of BnGlu NCA in analytical THF (55 mg/mL) at different time points.

Thus, our NCA won't be decomposed with the aid of PO/ECH and if we carry out the workup and purification steps appropriately (see SI). For example: if water wash step is involved, we usually conducted it at a low temperature and as quick as possible; the water-extracted organic phase need to be dried immediately with Na₂SO₄; for hydrophilic and unprotected NCAs, we never used water wash step in the workup; for rotavapor, we kept the temperature as low as possible (< 40 °C); for chromatography, regular solvents are fine but the silica gel were oven-

dried. Those precautions are all accounted for slowing down possible water-induced side reactions. Once purified, the NCAs still need to be stored at lower temperatures to minimize the moisture-triggered side reactions.

For the experiments comparing epoxide acid scavengers to pinene or “traditional” NCA synthesis, the experimental conditions are not well defined. How many equivs of pinene vs epoxide? What is “tradition” defined as? The reactions need to be compared with the same equivalents of pinene as epoxide. Most of the other monomers synthesized have been reported elsewhere. Some compounds have been reported in in hundreds of publications over many decades, and so speeding the reaction from 4 h to 1 h or dry vs wet solvent isn’t viewed as a game changer.

Our response: The equivalent of pinene was 4, the same as epoxide. The information is now available in the revised table (also shown below)

1: R² = R³ = H
2: R² = alkyl, R³ = Boc

3: R² = H
4: R² = alkyl

NCA	Yield with new method	Traditional method ^d	Yield with α-pinene ^e
 3p crystal of 3p	88% ^a	0%	0%
 3q crystal of 3q	46% ^a	0%	-
 4c 2.7 g	72% ^b	0%	-
 3r 11.5 g	78-95% ^c	0%	0%
 3s crystal of 3r	68% ^c	0%	-
 3t 10.8 g	92% ^a	0%	0%

^acondition B: 0.5 eq triphosgene, 4 eq ECH, THF, rt.

^bcondition A-2: 0.5 eq triphosgene, 4 eq PO, CH₃CN, 0 °C.

^ccondition A-1: 0.5 eq triphosgene, 4 eq PO, THF, rt.

^dcondition: 0.5 eq triphosgene, anhydrous THF, 50 °C.

^econdition: 0.5 eq triphosgene, 4 eq pinene, THF, rt.

While the substitution of an epoxide for pinene seems to be a nifty synthetic tip worthy of disseminating to those in the field, the innovation is not stark enough to warrant publication here. And if the true innovation is access to protecting group-free NCAs, this aspect should be expanded upon.

Our response: We respectfully disagree with the reviewer on these comments. We believe that this paper warrants publication in *Nat. Commun.* not just because the method that we developed shows robustness under moist conditions, but also because of the following advantages it confers (we would like to point out that these advantages were also recognized by both Reviewer 1 and 2 in general). First, our work provided new mechanistic insight into NCA synthesis and explained why previous acid scavengers had limited success (Figure 1 and Figure S3-4). Of note, the detrimental role of HCl in the formation of NCA has not been fully elucidated from a mechanistic perspective until this work. Second, the method detailed in this study allows synthesis of unprotected and challenging NCAs was made possible under mild conditions (Scheme 2). Third, the work also allows rapid access to important linear and novel hyperbranched polypeptides with higher yield, fewer steps, and more environmentally friendly procedures (Figure 2).

With the current PLG and branched polypeptides presented in the paper, I think it is sufficient to prove the utility and potential of the unprotected NCAs. It is unrealistic and out-of-scope to demonstrate all of the unprotected NCA in one single paper, particularly when NEW monomers are involved. However, we are happy to share with the reviewer that we are actively pursuing the ROP or Co-ROP of these new monomers. For example, more detailed mechanistic and characterization studies and biomedical applications of the hyperbranched polypeptides are ongoing. Using the ProNCA prepared by this method, we have recently demonstrated that we could achieve ultra-fast ROP of ProNCA and obtain high M_n poly-L-proline (preprint available in ChemRxiv. DOI: 10.26434/chemrxiv.14663883.v1). Copolymerizations of ProNCA with HypNCA, SerNCA, or GluNCA are also undergoing. We expect to report those results in several papers in the near future.

Reviewers' Comments:

Reviewer #2:

Remarks to the Author:

I checked the revised manuscript and the response to the referee's comments. I was fully convinced and I can recommend this manuscript for publication in Nat. Commun.

Reviewer #3:

Remarks to the Author:

I commend the authors on the additional effort and experiments in this revision, and the plethora of data in general. Overall, this reviewer finds that the manuscript hits the surface of many very important challenges such as mechanistic studies, acid scavengers, protecting group free polymer synthesis, and brush polymers. But regretfully I do not support publication in the current form because the work is still scattered and doesn't provide enough data to tell a complete story on any of these topics, and the claims overreach the data.

The authors have now directly stated the proposed innovations of this work as numbered below.

1. Providing mechanistic insight into, and elucidating the role of HCl, in NCA cyclization reactions.

The detrimental role of HCl seems to be well known since a quick literature search turned up studies on acid scavengers for NCAs spanning decades (pinene, TEA, activated carbon, celite, polymer resins, silica, aqueous workup). Page 4 mentions some of these but doesn't compare yields or convenience of each procedure. In looking at some of those papers the NCA yields are similar or even higher than those in scheme 2. The computational modeling indeed explains the rate acceleration with epoxide, but only 1 sentence in the whole paper is dedicated to Fig 1C. Amine vs acid paths are buried in the SI, and calculations are only performed for alanine. What about something very different like Pro or Sar? The calculations don't go into depth and don't seem to be a focus.

2. Synthesis of unprotected and challenging NCAs

3. Access to linear and hyperbranched polypeptides with higher yield, fewer steps, and more environmentally friendly procedures

Avoiding the use of protecting groups to save time/yield/environmental impact is fantastic and highly innovative. However, the monomers must be able to be polymerized into useful polymers. With so little polymer data it's not clear if their cyclization method provides this. Therefore, the data doesn't fully support the stated innovations 2,3. The authors said in the rebuttal it's unrealistic and out of scope to polymerize the monomers they prepared. But it was apparently realistic to prepare all the monomers shown in Scheme 2, so why not provide a little more data than a couple PGA (PLG?) polymers? No data is provided for unprotected polySer, polyThr, polyTrp, polyTyr, the phosphoTyr etc. Expansive studies need not be done, but so little data is provided that broad access to unprotected polypeptides can't be claimed.

In the previous revision I also noted the author's claim of controlled polymerization of Glu NCA was not fully supported. Controlled polymerization can be established by demonstrating linear chain growth over a range of molecular weights (MWs). Only 2 data points for low MW polymers was included so I requested needed data. However, it is still not provided. The authors should scale back their claims until such data is provided. They report up to ~40 mers, but what lengths are typically used in the medical applications cited. Are these long enough?

The authors did supply a new chain extension experiment to show the living character, which is helpful. There is, unfortunately, some discrepancy in the new NMR data for the material. The methoxy methyl group is calibrated at 3 protons. The protons for b, b, c, c' should then integrate to 8 but there are only 5.6 protons. The peaks for d, e should be 4 protons but there are 11.1.

Point-to-Point Response

REVIEWERS' COMMENTS

Reviewer #2 (Remarks to the Author):

I checked the revised manuscript and the response to the referee's comments. I was fully convinced and I can recommend this manuscript for publication in Nat. Commun.

Our response: We gratefully thank reviewer 2 for all the comments and supports.

Reviewer #3 (Remarks to the Author):

I commend the authors on the additional effort and experiments in this revision, and the plethora of data in general. Overall, this reviewer finds that the manuscript hits the surface of many very important challenges such as mechanistic studies, acid scavengers, protecting group free polymer synthesis, and brush polymers. But regretfully I do not support publication in the current form because the work is still scattered and doesn't provide enough data to tell a complete story on any of these topics, and the claims overreach the data.

Our response: We appreciate the comments of reviewer 3 that have helped us improved our paper significantly. We have examined our manuscript thoroughly and removed all exaggerating words. All changes can be tracked from the file entitled "manuscript with changes highlighted"

The authors have now directly stated the proposed innovations of this work as numbered below.

1. Providing mechanistic insight into, and elucidating the role of HCl, in NCA cyclization reactions.

The detrimental role of HCl seems to be well known since a quick literature search turned up studies on acid scavengers for NCAs spanning decades (pinene, TEA, activated carbon, celite, polymer resins, silica, aqueous workup). Page 4 mentions some of these but doesn't compare yields or convenience of each procedure. In looking at some of those papers the NCA yields are similar or even higher than those in scheme 2. The computational modeling indeed explains the rate acceleration with epoxide, but only 1 sentence in the whole paper is dedicated to Fig 1C. Amine vs acid paths are buried in the SI, and calculations are only performed for alanine. What about something very different like Pro or Sar? The calculations don't go into depth and don't seem to be a focus.

Our response: The reviewer is absolutely right on the point that the NCA field has long

recognized the beneficial fact of removal of HCl. The intuitive motivation, however, is simply providing more thermodynamic driving force for the cyclization in most cases.

The new information uncovered in this work from a mechanistic point of view, however, is that we showed how fast NCA decompose or polymerize under different conditions through modal kinetic reactions. We also investigated through DFT the different phosgenation routes. The results offered important understanding on the subtle relationship of the NCA-HCl-H₂O trio system, which we believe has been largely overlooked by most researchers in the field. The results explained why most NCAs synthesis has to be conducted under stringent anhydrous conditions previously, and laid the foundation of our hypothesis regarding performing NCA synthesis in water-containing solvents, as long as HCl can be instantaneously removed from the system before it could catalyzes NCA decomposition. We have rewrote this part in the Discussion section to more precisely describe our mechanistic studies.

“The effect of HCl-removing in the formation of NCA has been partially recognized by the field, but its detrimental role has not been fully elucidated from a mechanistic point of view. Acid scavengers, such as TEA and α -pinene, have been used as additives and achieved certain successes in the synthesis of some, but not all, NCA monomers. In Figure 2a, we demonstrated that the stability of NCA towards water is highly dependent on the presence of HCl. We therefore propose that NCA synthesis under mild, atmospheric conditions, without the need for moisture removal could be achievable as long as HCl could be efficiently removed from the crude products before the HCl-catalyzed NCA decomposition taking place. After ruling out TEA and α -pinene, we identified PO and ECH as ultra-fast and clean HCl-quenching reagents suitable for NCA preparation.”

For the NCAs that are easy to make, our method offered comparable or better yields and simpler workup than previous methods in most examples. I believe these results are acceptable if one consider that the most important contribution of this method is the abandon of dry solvents, glove box, and enabling the synthesis of challenging NCAs.

Alanine is used as a model monomer in the DFT study for its relative simple structure and thus less amount of time in calculation. The goal of this study is to provide mechanistic insights from a representative example rather than exhausting all monomers. Although the syntheses of SarNCA and ProNCA were slightly different from other monomers such as AlaNCA, in principle, it doesn't change how the addition of PO/ECH affect the NCA formation. Therefore, we respectfully declined the suggestion of performing more DFT studies for Sarcosine and proline. Per the reviewer's request, we have added some discussion about Figure 1C (now figure 2C) in the Discussion section:

“In addition to the above-mentioned benefits, the rapid removal of HCl by PO/ECH also allowed accelerated preparation of NCA without heating via lowering the

cyclization energy barrier (Figure 2c), which is important for the synthesis of some labile or highly reactive monomers.”

2. Synthesis of unprotected and challenging NCAs

3. Access to linear and hyperbranched polypeptides with higher yield, fewer steps, and more environmentally friendly procedures

Avoiding the use of protecting groups to save time/yield/environmental impact is fantastic and highly innovative. However, the monomers must be able to be polymerized into useful polymers. With so little polymer data it's not clear if their cyclization method provides this. Therefore, the data doesn't fully support the stated innovations 2,3. The authors said in the rebuttal it's unrealistic and out of scope to polymerize the monomers they prepared. But it was apparently realistic to prepare all the monomers shown in Scheme 2, so why not provide a little more data than a couple PGA (PLG?) polymers? No data is provided for unprotected polySer, polyThr, polyTrp, polyTyr, the phosphoTyr etc. Expansive studies need not be done, but so little data is provided that broad access to unprotected polypeptides can't be claimed.

Our response: The main focus of this paper is the new method of NCA preparation, which could be used by the researchers in both academia and industry for the easier synthesis of polypeptides and related materials. To demonstrate that the NCAs prepared by our method are pure and polymerizable, we have provided sufficient polymerization results in Supplementary Table 2, Supplementary Figure 25, and Figure 5. We believe expansive ROP studies will distract readers and make the paper out of focus, given the fact that the paper is already a bit lengthy. On the other hand, based on my understanding of the policy of *Nature Communications*, all of our corresponding including this response letter will appear online to public. For this reason, even though we do have accumulated some very exciting copolymerization results of SerNCA, HypNCA, and PenNCA, we would like to reserve them for a number of publications in the next 1-2 years. We should point out once again that the homopolymerizations of for NCAs such as Ser, Thr, Tyr, and Phe, are rarely done because their homopolypeptides are insoluble in most organic solvents. Finally, we feel the reviewer seems oversimplified the ROP of NCAs, which could have dramatically different reactivity and behaviors depending on the side groups. For example, as we shared in our second response letter, we recently reported the water-assisted controlled ROP of ProNCA (preprint available in ChemRxiv. DOI: 10.26434/chemrxiv.14663883.v1). We could appreciate from this study that even for a seemingly well-known and simple NCA, interesting new insights and better polymerization methods can still be implemented after decades of its invention, not to mention those novel and reactive NCAs.

Overall, the major reason of including the ROP results of GluNCA and CysNCA is to prove these new NCAs are pure and polymerizable. With the current results, it shouldn't take much thinking for our readers to realize the huge potential of these new monomers in creating fascinating new materials.

In the previous revision I also noted the author's claim of controlled polymerization of Glu NCA was not fully supported. Controlled polymerization can be established by demonstrating linear chain growth over a range of molecular weights (MWs). Only 2 data points for low MW polymers was included so I requested needed data. However, it is still not provided. The authors should scale back their claims until such data is provided. They report up to ~40 mers, but what lengths are typically used in the medical applications cited. Are these long enough?

The authors did supply a new chain extension experiment to show the living character, which is helpful. There is, unfortunately, some discrepancy in the new NMR data for the material. The methoxy methyl group is calibrated at 3 protons. The protons for b, b, c, c' should then integrate to 8 but there are only 5.6 protons. The peaks for d, e should be 4 protons but there are 11.1.

Our response: different M_n of PLG can all find potential applications. For example, as we mentioned in our second revision letter, Prof. Kataoka have used PEG-PLG block copolymer as carriers to deliver various drugs, and those technologies have been pushed forward to nanomedicines in different stages of clinical trials (e.g. NC-6004 and others, NanoCarrier Ltd.). The DP of PLG in his systems is in the range of 15-30 (*Nat. Nanotechnol.* **2011**, *6*, 815-823. (DOI: 10.1038/nnano.2011.166)). So the two PLG we prepared are long enough for such drug delivery applications.

We sincerely thank Reviewer 3 for identifying the discrepancy of peak integration of our NMR spectrum (Supplementary Figure 34). We believe this discrepancy is possibly due to: (1) unideal shimming and incorrect baseline processing in the Mestrenova software; (2) the hydrogen bonding and alpha-helix of P(EG₃Glu) make the NMR integration inaccurate, which has been very often in other polypeptides. For this, we have retaken the NMR spectrum of the same sample in D₂O/NaOD and D₂O/TFA-*d*. Both spectra are available in Supplementary Figure 34 (also shown below). As we can tell, both NMR spectra confirmed a roughly 1/1 ratio of PLG/P(EG₃Glu).

^1H NMR spectra of PLG-*block*-P(EG₃-Glu) in (A) NaOD/D₂O (B) TFA-*d*/D₂O